# Contribution of atmospheric circulation to recent off-shore sea-level variations in the Baltic Sea and the North Sea

Sitar Karabil, Eduardo Zorita, Birgit Hünicke

Institute of Coastal Research, Helmholtz-Zentrum Geesthacht. Max-Planck Str.1, Geesthacht, 21502, Germany

*Correspondence to*: Sitar Karabil (starkarabil@gmail.com)

**Abstract.** The main purpose of this study is to quantify the contribution of atmospheric factors to recent off-shore sea-level variability in the Baltic Sea and the North Sea on interannual time scales. For this purpose, we statistically analysed sea-level records from tide gauges and satellite altimetry and several climatic data sets covering the last century.

Previous studies had concluded that the North Atlantic Oscillation (NAO) is the main pattern of atmospheric variability

affecting sea-level in the Baltic Sea and the North Sea in wintertime. However, we identify a different atmospheric circulation pattern that is more closely connected to sea-level variability than the NAO. This circulation pattern displays a link to sea-level that remains stable through the 20[th] century, in contrast to the much more variable link between sea-level and the NAO. We denote this atmospheric variability mode as the Baltic Sea and North Sea Oscillation (BANOS) index. The sea-level-pressure (SLP) BANOS pattern displays an SLP dipole with centres of action located over (5° W, 45° N) and (20°

E, 70° N) and this is distinct from the standard NAO SLP pattern in wintertime. In summertime, the discrepancy between the SLP BANOS and NAO patterns becomes clearer, with centres of action of the former located over (30° E, 45° N) and (20° E, 60° N).

This index has a stronger connection to off-shore sea-level variability in the study area than the NAO in wintertime for the period 1993-2013, explaining locally up to 90% sea-level of the interannual sea-level variance in winter and up to 79% in

summer. Sea-level in the eastern part of the Gulf of Finland is the most sensitive area to the BANOS-index in wintertime, whereas the Gulf of Riga is the most sensitive region in summertime. In the North Sea region, the maximum sea-level sensitivity to the BANOS pattern is located in the German Bight for both winter and summer seasons.

We investigated, and when possible quantified, the contribution of several physical mechanisms which may explain the link between the sea-level variability and the atmospheric pattern described by the BANOS-index. These mechanisms include the

inverse barometer effect (IBE), fresh water balance, net energy surface flux and wind-induced water transport. We found that the most important mechanism is the IBE in both wintertime and summertime. Assuming a complete equilibration of seasonal sea-level to the SLP gradients over this region, the IBE can explain up to 88% of the sea-level variability attributed to the BANOS-index in wintertime and 34% in summertime. The net energy flux at the surface is found to be an important factor for the variation of sea-level, explaining 35% of sea-level variance in wintertime and a very small amount in summer.

The freshwater flux could only explain 27% of the variability in summertime and a negligible part in winter. In contrast to

the NAO, the direct wind forcing associated to the SLP BANOS pattern does not lead to transport of water from the North Sea into the Baltic Sea in wintertime.

**Keywords:** off-shore sea-level, atmospheric factors, the Baltic Sea, the North Sea, statistical analysis.

## 1 Introduction

Variations of regional sea-level can deviate substantially from the globally averaged sea-level (Church et al., 2013), due to the diversity of regional driving factors that may affect sea-level variations.

The Baltic Sea and its interconnection to the North Sea have been widely investigated for a better understanding of sea-level variability in this region (e.g. Yan et al., 2004; Novotny et al., 2006; Passaro et al., 2015). It is known that sea level variations in the Baltic Sea and the North Sea from interannual to multidecadal time scales are strongly driven by the

atmospheric circulation, in particular by the North Atlantic Oscillation (NAO) (e.g. Andersson, 2002; Hünicke and Zorita, 2006; Dangendorf et al., 2012). The NAO is a mode of the large-scale atmospheric circulation in wintertime that dominates the atmospheric variability over Europe and the North Atlantic from interannual to decadal time scales (e.g. Hurrell 1995; Osborn et al., 1999; Hurrell et al., 2003; BACC II, 2015).

The NAO represents the anticorrelation between sea-level-pressure (SLP) over the northern North Atlantic, centred over

Iceland, and the subtropical high-pressure cell centred over the Azores. The variations of the meridional SLP gradient are linked to the strength of the mean winter westerly winds over northern Europe and to the advection of oceanic air masses into this continent (Hurrell, 1995; Hurrell et al., 2003b; Slonosky et al., 2000). The temporal variations of the NAO can be described by the NAO-index, which can be constructed from the differences in two normalised SLP records, such as between Azores high and Icelandic low (Hurrell, 1995; Jones et al., 1997). Normalisation is needed to filter the series being

dominated by the larger variability of the northern station with respect to variability of the southern station (Hurrell et al., 2003).

The phases of the NAO have different influences on the climate of northern Europe. For instance, on the one hand, a positive phase of the NAO-index is associated with strong westerly winds transporting warm humid air masses eastward and resulting in mild winters over the northern Europe including the Baltic Sea (e.g. Hurrell, 1995; Hurrell et al., 1997; Hurrell et

al., 2003). A negative phase of the NAO-index describes weaker westerly winds or even westward advection of cold and dry Siberian air towards Europe (e.g. Hurrell, 1995; Hurrell et al., 2003; Hagen and Feistel, 2008).

In general, a positive (negative) phase of the NAO causes sea-level to rise (fall) in the Baltic Sea and the North Sea (e.g. Wakelin et al., 2003; Hünicke and Zorita, 2008; Dangendorf et al., 2012; Hünicke et al., 2015). The NAO may directly impact on sea-level variations in the Baltic Sea and the North Sea in several ways.

The NAO-related westerly winds can transport water into the Baltic Sea from the North Sea basin through the transition zone (e.g. Kauker and Meier, 2003; Ekman, 2009). Another related mechanism is that NAO influences the temperature of northern Europe including the Baltic Sea. Variations in temperature can affect the sea-level due to thermal expansion of the

water column (e.g. Hünicke and Zorita, 2006; Dangendorf et al., 2012). A third possible mechanism involves the modifications of the surface water balance, which can affect the sea level-variability in a semi-enclosed sea like the Baltic Sea. For instance, a positive phase of the NAO may cause a positive fresh water balance resulting primarily from the higher precipitation within the Baltic Sea drainage basin (e.g. Hünicke and Zorita, 2006; Hünicke et al., 2008; Lehmann et al., 2011). These mechanisms can not only change the total water volume of the Baltic Sea, but also the water density through changes in salinity.

Additionally, it is also reasonable to expect an influence of pressure differences on sea-level variability due to the NAO-related large-scale changes in the SLP field through the inverse barometer effect (e.g. Yan et al., 2004). There can be NAO-associated indirect factors such as snow and ice melt, river run-off affecting the sea-level variations in this region as well.

Overall, these mechanisms make the NAO important in order to describe the effect of atmospheric forcing on sea-level variability, especially in wintertime (e.g. Jevrejeva et al., 2005; Ekman, 2009; Stramska and Chudziak, 2013).

However, it has been found that the impact of the NAO on sea-level varies substantially across the Baltic Sea and the North Sea (e.g. Hünicke and Zorita, 2006; Tsimplis and Shaw, 2008). For example, (Yan et al., 2004) investigated the relationship between the NAO and the sea-level around northern European coasts and reported that the NAO is positively correlated with sea-level variations on annual and longer time scales along the central and northern coasts of the Baltic Sea and the German Bight. They found that these positively correlated regions display higher sea-level under the stronger NAO phases, more clearly so in wintertime, whereas non-significant and even negative relationships to the NAO exist over the southern coast of the Baltic Sea and southwest England, respectively. They also concluded that the link between sea-level variations and the NAO is variable in time and has increased over both negatively and positively linked regions over the last decades. These findings are also consistent with results of other studies such as Andersson (2002), Wakelin et al. (2003), Jevrejeva et al. (2005) and Hünicke and Zorita (2006). Furthermore, several studies including Andersson (2002) who focused on the Baltic Sea, and Dangendorf et al. (2014) who investigated the southeastern North Sea, reported that atmospheric variability that may be described by patterns different from the NAO may still explain part of the sea-level variability. Some previous studies also showed that, on average, the NAO accounts for only one-third of total sea-level variability in the Baltic Sea on the interannual time scale (e.g. Kauker and Meier, 2003; Jevrejeva et al., 2006).

Using the Stockholm tide gauge and the NAO-index time series, Andersson (2002) analysed the influence of atmospheric circulation on the Baltic Sea level. Her study indicated that interannual sea-level variations, particularly along the northern and eastern coasts, are highly modulated by the strong westerlies related to the NAO. In addition, she suggested that an atmospheric circulation index (the BAC-index) which can be constructed from the difference between normalised time series of the air pressure centres closer to the Baltic Sea entrance can describe the winter mean of Baltic Sea level variations better than the NAO. Related to this suggestion, Novotny et al. (2006) found a substantial correlation between air-pressure over the North Sea and interannual sea-level variations in the Baltic Sea.

Concerning other sorts of data sets including the satellite altimetry observations, there have been several studies focused on long-term (interannual and longer) sea-level variations in the Baltic Sea and its connection to the North Sea. Stramska (2013)

investigated off-shore and coastal sea-level variability in the Baltic Sea and the North Sea. Using the satellite altimetry observations, tide gauge records and meteorological data sets, she confirmed the high coherence of sea-level variability between the Baltic Sea and the North Sea. By exploring the sea-level variability in the Baltic Sea on interannual and decadal time scales, Xu et al. (2015) demonstrated that altimeter data have high correlations with tide gauge data in these regions,

except in the Danish straits. They also showed that the basin-averaged altimetric Baltic Sea level exhibits strong correlation with the NAO-index in wintertime. Using Envisat altimetry data together with tide gauge observations in the Baltic Sea-North Sea transition zone covering the Danish straits, Passaro et al. (2015) concluded that coastal altimetry is able to capture the annual sea-level variations on a sub-basin scale.

Most of those previous studies addressed the link between coastal sea-level variability and atmospheric forcing, identifying

the NAO as the most relevant atmospheric pattern for sea-level variability in the Baltic Sea. However, the link between the NAO and the Baltic Sea level is known to be unstable in time and quite heterogeneous in space. The correlation between sea-level records and the NAO calculated over gliding multidecadal windows in the 20th century displays periods in which this correlation is very high, of the order of 0.8 for some tide gauge records like in the most recent two decades, but also periods in which this correlation is as low as 0.3 and even may turn negative for some tide-gauges located in the southern Baltic Sea

(e.g. Hünicke and Zorita, 2006).

In this study, we revisit the link between the atmospheric circulation and sea-level variability in the Baltic Sea and the North Sea with the aim of ascertaining whether the NAO is indeed the most relevant pattern and whether there may be other atmospheric patterns that display a stronger and more stable connection in time with sea-level variations in this region. This analysis leads us to describe a new index of atmospheric circulation that we denote the Baltic Sea and North Sea Oscillation

(BANOS) index. We also analyse other meteorological fields, like surface energy fluxes, to investigate the physical mechanisms that may explain the connection between the BANOS pattern of atmospheric circulation and the Baltic Sea and North Sea sea-level. The identified BANOS pattern bears some resemblance to another mode of atmospheric circulation that has been found to be connected to sea-level variations in Cuxhaven, at the German North Sea coast (Dangendorf et al., 2014). Whereas the mechanism for sea-level variations at this part of the North Sea are related to direct wind forcing, we

find different physical mechanism to be responsible for the link to the Baltic Sea level variability.

As mentioned before, although the direct wind forcing has been assumed to be the main factor explaining this connection, there are other candidates that may also be involved. Since the link between the NAO and sea-level varies considerably in time, it is reasonable to assume that other physical mechanism may also contribute to sea-level variability.

The present analysis is not restricted to the coastal sea-level and makes use of satellite altimetry data to obtain a more

complete picture of the link between atmospheric circulation and sea-level variability, using the most recent available altimetry data set, which is interpolated from along-track observations onto a 1/4°x1/4° resolution, over the Baltic Sea and North Sea region and extended to the North Atlantic in order to reveal possible large-scale effects on the off-shore sea-level variability.

A map showing the study area with an overview of some basins and subdivisions of the Baltic Sea and the North Sea is

represented in Figure 1. More subdivisions can be defined in terms of focus and spatial scale of the study. Due to the scope of this study, we included some and labelled them in Figure 1.

**Figure 1: The Study Area with sub-regions. 1-Bothnian Bay, 2-Bothnian Sea, 3-Gulf of Finland, 4-Baltic Proper, 5-Gulf of Riga, 6-Arkona Basin, 7-Danish Straits, 8-Kattegat, 9-Skagerrak, 10-North-eastern North Sea, 11-German Bight. The study area is also shown together with the tide gauge locations: Helsinki, Sassnitz, Warnemünde, Wismar, Travemünde, Smögen, Kungsholmsfort, Stockholm and Ratan (names are written in an order starting from far east station and follows clockwise rotation).**

This study focuses on the winter (December-to-February) and summer (June-to-August) seasons.

This paper is organised as follows: Section 2 presents the data sets used in this study and the following section describes the applied statistical methods. Section 4 includes the main results of this study. In the discussion and conclusions section, we assess the results.

## 2 Data Sets

The study uses primarily two different types of data sets. The first set consists of monthly means of sea-level observations obtained from satellite altimetry missions and tide gauges. We prescribed a minimum threshold of 75% data availability to include a particular time series in the analysis. However, we computed seasonal means of a tide gauge record if two months within one season were available. The second set comprises climatic data including sea-level-pressure (SLP) observations and the NAO-index. The climatic data set also contains meteorological reanalysis data of precipitation, surface fluxes including short-wave and long-wave radiative fluxes, sensible and latent heat turbulent fluxes.

In this study, we used seasonal mean values for winter and summer seasons. Those winter and summer mean values are computed from monthly means prior to the analysis. We analysed winter (December-January-February) and summer (June-July-August) seasons separately. Therefore, there was no need to remove the seasonal cycle for our analysis.

The satellite altimetry observations - Sea Level Anomalies (SLAs) - are retrieved from the delayed time multimission global gridded data products provided by the AVISO (www.aviso.altimetry.fr). We used the state-of-the-art satellite altimetry data interpolated onto a 1/4°x1/4° cartesian grid resolution (DT2014 SLA) with 20-year mean reference period. More information about the data can be found in AVISO/CNES product report published in April 2014. For our study, we considered winter and summer seasons, covering the period from December 1992 to August 2013 for the defined geographical window between (45° W - 30° E) longitudes, and (48° N - 70° N) latitudes.

Most of the tide gauge records representing monthly mean sea-level variations are provided by the Permanent Service for Mean Sea-Level (PSMSL - www.psmsl.org) (Woodworth and Player, 2003). The largest part of the Stockholm tide gauge record was provided by Ekman (2003). Some parts of the Sassnitz, Travemünde, Warnemünde and Wismar tide gauge observations have been provided by Technische Universität Dresden. Overall, including the northeast boundary of the North Sea, nine tide gauges along the Baltic coasts were selected: Ratan, Helsinki, Stockholm, Smögen, Kungsholmsfort, Sassnitz, Travemünde, Wismar and Warnemünde. The selection was based on their geographical distribution and the record length of the tide gauges. The Smögen station has some missing values at the beginning of the analysis period; the other tide gauges

provide complete records for the whole analysis period 1900-2013.

We considered four different data sets concerning climatic variables.

The sea-level-pressure (SLP) data are monthly means of 5°x5° gridded observations covering the Northern Hemisphere, from 1900 to 2013, provided by the National Centre for Atmospheric Research (https://climatedataguide.ucar.edu/climate-data/ncar-sea-level-pressure) (Hurrell et al., 2016a). The geographical domain of the data set used here is (70° W - 40° E) and (30° N - 90° N).

The North Atlantic Oscillation (NAO) is a pattern of the large-scale pressure fields over the North Atlantic region. The time-varying intensity of this pattern can be summarised by the NAO-index. The NAO-index that we used was computed by Hurrell et al. (2016b) using the differences between normalized anomalies of two SLP stations: Lisbon, in Portugal and Reykjavik, in Iceland (https://climatedataguide.ucar.edu/climate-data/hurrell-north-atlantic-oscillation-nao-index-station-based).

We used the National Center of Environmental Prediction and National Center for Atmospheric Research (NCEP/NCAR) reanalysis data covering the period from 1949 to 2013 (Kalnay et al., 1996; Kistler et al., 2001) (https://www.esrl.noaa.gov/psd/data/gridded/data.ncep.reanalysis.surfaceflux.html). The meteorological reanalysis assimilates different observations into a weather prediction model and produces as a complete gridded data set over the whole period.

This data set has spatial resolution of a T62 Gaussian grid with non-regular 192x94 points covering the Earth's surface and is available as 4-times daily, daily and monthly values from 1948/01/01 to present. Here, we considered the monthly means of surface fluxes of net short-wave radiation, net long-wave radiation, net sensible, net latent heat fluxes and precipitation.

To investigate the possible strong connection between sea level variability in this region and other atmospheric circulation patterns, we have also measured the strength of the relation between the Stockholm, Warnemünde and Cuxhaven tide gauges and three local teleconnection patterns. These patterns were previously suggested by (Barnston and Livezey, 1987) and their time series are available through the server of National Oceanic and Atmospheric Administration, Climate Prediction Center (NOAA-CPC), (http://www.cpc.ncep.noaa.gov/data/teledoc/telecontents.shtml).

**3 Methods**

The present investigation is based on statistical analysis of sea-level and climatic data. The methods used in this study are widely known, but still summarised here for the sake of completeness.

We computed winter and summer means from December to February and from June to August monthly means, respectively. Moving correlations are computed over gliding 21-year periods for winter means and of summer means separately. The sea-level means and climate records are linearly de-trended prior to the related statistical analysis to filter out the effect of secular sea-level rise and the effect of land-crust movements that are not related to the atmospheric circulation. In particular, the Baltic Sea records display a clear long-term trend that is in part caused by the crust rebound after the last deglaciation

(Glacial Isostatic Adjustment).

In addition, we estimated the sensitivity values of sea-level variations based on one unit change in the atmospheric circulation index by means of a linear regression analysis with the sea-level records as predictands and the atmospheric indices as predictors. The linear regression parameter of the regression analysis is denoted as the sensitivity of sea-level to

changes in the intensity of the atmospheric patterns. We further used this method to explore the physical mechanism explaining the connection between indices of atmospheric circulation and sea-level in addition to the correlation analysis. This approach is used in different studies to have a better understanding of the statistical linkage between atmospheric condition and sea-level variation in this region (e.g. Wakelin et al., 2003; Dangendorf et al., 2012; Chen et al., 2014). It should also be noted that the estimation of sensitivity values improves the understanding of the connection between

atmospheric circulation and sea level, since the sensitivity values of sea level to the atmospheric index may differ in a case that correlation values between those sea level variations and atmospheric circulation indicate the same covariation. This helps identify more vulnerable off-shore and coastal locals in responding to the atmospheric circulation.

In this manuscript, the term "coherent" indicates only high degree of covariation and does not necessarily provide information about the relative variation of two variables.

**4 Results**

**4.1 Comparison between satellite SLAs and tide gauges**

We first examine how the satellite altimetry observations (SLAs) covary with the tide gauge records. For this examination, we selected the individual grids closest to each of the nine tide gauges separately. The de-trended 21-year period time series for the period 1993-2013 are used to calculate the correlation coefficients displayed in Table 1.

**Table 1: The correlations between individual satellite altimetry grids and the tide gauges for the period 1993-2013.**

As the high correlations indicate, satellite altimeter observations are found to be coherent with the tide gauges. These results also indicate a substantial progress in satellite altimetry when we consider earlier the results from the study of Yan et al. (2004). In that study, for example, the correlation coefficient between the closest SLA grid cell to Wismar and the Wismar tide gauge was ~0.50. This good agreement is likely due to the updating of geophysical corrections and usage of refined

mapping parameters. Further information on the improvement of DT2014 SLA satellite altimeter data is provided by Pujol et al. (2016).

After testing the coherency of satellite altimetry observations with respect to the tide gauge records, we calculated the correlation coefficients between each of the nine tide gauges (Ratan, Stockholm, Helsinki, Smögen, Kungsholmsfort, Travemünde, Warnemünde, Wismar, Sassnitz) and all available SLA grid-cells. These correlation patterns show how the

coastal sea-level variations at the tide gauges are linked to open sea-level variability. Figure 2 displays the obtained correlation patterns for the winter season.

**Figure 2: The correlation patterns between the SLA grids and tide gauge records in wintertime (DJF) for the period 1993-2013. The value of correlation significance is ±0.43 at the 95% confidence level for this record length.**

Figure 2 indicates that satellite altimetry time series are strongly correlated to tide gauges in the whole Baltic Sea and in large parts of the North Sea in wintertime over the period 1993-2013. Only the northern North Sea basin results in weak correlation values. The North Atlantic shows negative correlations to the Baltic Sea tide gauges. It should be noted that the seesaw pattern suggests negative and positive dipole relation between the North Atlantic Ocean and the Baltic Sea basin. Elsewhere, the correlation patterns are spatially homogeneous over the Baltic Sea and North Sea region. It should be noted that the relation between the SLA data and the tide gauge records becomes relatively weaker for the southern Baltic Sea stations (Travemünde, Wismar and Warnemünde). Considering only the North Sea, the correlations between tide gauges and SLA display an increasing pattern from west to east. The maximum correlation of these patterns is found in the German Bight (r=0.94). At the following, the correlation patterns for the summer season are displayed in Figure 3.

**Figure 3: The correlation patterns between the SLA grids and tide gauge records in summertime (JJA) for the period 1993-2013. The correlation coefficients are computed based on de-trended seasonal means between SLA grids and the tide gauge records. The value of correlation significance is ±0.43 at the 95% confidence level for this record length.**

In general, the correlation patterns display a strong (max. r~0.92) link between the satellite altimetry and tide gauges over the whole Baltic Sea in summertime. Additionally, the patterns of correlations exhibit a quite uniform spatial distribution in this region. For example, when we consider the correlation patterns of the tide gauges Ratan, Stockholm, Helsinki and Kungsholmsfort, the correlation is found to be the strongest over the Bothnian Sea, but it is slightly weaker in the northern Baltic proper and in the Bothnian Bay. This strong correlation pattern extends as far west as Skagerrak. In the North Sea, however, only the area between north-eastern North Sea and the German Bight displays high correlations for all tide gauges in summertime.

## 4.2 Relationships between the NAO-index and satellite SLAs

In this sub-section, we first correlated the SLAs obtained from the satellite altimeter observations with the NAO-index for the period 1993-2013. The correlation patterns of winter and summer seasons are displayed in Figure 4. To detect the possible large-scale impact on the off-shore sea-level variability in the Baltic Sea and the North Sea, we also included a part of the North Atlantic.

**Figure 4: The correlation maps between the SLA grids and the NAO-index for the winter – means of DJF months - (top) and summer – means of JJA months - (bottom) seasons (1993-2013). The correlation coefficients are computed based on de-trended seasonal means between SLA grids and the NAO-index. The value of correlation significance is ±0.43 at the 95% confidence level for this record length. The areas indicating significant correlations are delineated with contour lines.**

In wintertime, the correlation pattern between SLA grids and the NAO-index seems to be spatially uniform (r~0.71) over the entire Baltic Sea and the major part of the North Sea. In particular, the German Bight has a relatively stronger connection to the NAO than the rest of the North Sea. This agrees with previous studies (Wakelin et al., 2003; Woolf et al., 2003; Dangendorf et al., 2012; Chen et al., 2014; Xu et al., 2015; Sterlini et al., 2016). The transition zone between the Baltic Sea and the North Sea has a spatially heterogeneous correlation with the NAO-index, with a maximum correlation of 0.55 and a

minimum of 0.01. The Baltic Sea displays a rather uniform correlation with the NAO-index and the maximum correlation occurs in the Bothnian Bay (r~0.76).

In summertime, there is almost no relation between the NAO-index and the SLAs in the Baltic Sea. Considering the North Sea, a part of the German Bight appears to be connected (r~0.60) to the NAO. However, the rest of the North Sea does not indicate a connection between sea-level and the NAO.

For this record length, the value of significance is ±0.43 at the 95% confidence level, under the usual assumptions of normally distributed and temporally uncorrelated variables. The significant correlation (r>0.6) between the NAO and the German Bight sea-level in summertime seems to contradict previous studies since they did not find a significant correlation between the NAO and the German Bight sea-level (e.g. Dangendorf et al., 2012). Additionally, it should be noted that tide gauge observations indicate a weak correlation to the NAO in the southern Baltic Sea in wintertime (e.g. Yan et al., 2004; Hünicke and Zorita, 2006). The associated weak correlation between tide gauges in the southern Baltic Sea and the NAO in wintertime contradicts the results obtained by SLAs here as well. Since those associated studies used the different period of sea-level records, the reason for those contradictory results could be that the connections between the NAO and the southern Baltic sea-level variability in wintertime and between the NAO and a part of the German Bight in summertime may not be stationary over time.

As pointed out, the patterns of correlation between the tide gauges and the satellite altimetry fields indicate that the variations of sea-level in these regions are spatially quite uniform in both seasons (Figure 2 and Figure 3) on the interannual time scale. However, the NAO-index does not seem to be strongly connected to the SLA grids for the same period in summertime (Figure 4-bottom).

## 4.3 Time evolution of the links between the NAO and the Baltic-North Sea levels

The results of the previous sub-sections show that, on the one hand, sea-level variations in the Baltic Sea and the North Sea tend to be spatially well correlated over the whole area. On the other hand, the influence of the NAO on sea-level in these areas is, in contrast, spatially quite heterogeneous (i.e. Yan et. al., 2004; Hünicke and Zorita, 2006). In addition to that, the connection between the NAO mode of atmospheric circulation and sea level variability in this region has seasonally quite heterogeneous patterns. These apparent contradictions would put into question that the NAO pattern is the major driver of sea-level variations in the Baltic and the North Seas. In this sub-section, we explore whether another pattern of atmospheric circulation, different from the NAO, can more strongly affect the sea-level variations over the entire Baltic Sea and the connection to the North Sea, both in the winter and summer seasons. At this stage, we investigated which patterns of atmospheric circulation might be more strongly related to sea-level variations on the interannual time scale than the NAO pattern. For this investigation, we first analysed the temporal stability of the statistical link between tide gauges and the NAO-index over the longer period 1900-2013.

The temporal stability of the correlation between the NAO and sea-level over the study area is examined based on the running correlations over 21-year windows between nine tide gauges and the NAO-index (not shown). The results indicate

that there is a temporal variability in the strength of the connection between tide gauges and the NAO, as already found by previous studies, i.e. Andersson (2002), Yan et al. (2004) and Hünicke and Zorita (2006).

To identify which atmospheric pattern may be more closely connected to sea-level variability and at the same time display a stable link over time, we selected two representative stations (Stockholm and Warnemünde) from nine tide gauges and
computed the 21-year moving correlations with the NAO over the years 1900 - 2013. The results are plotted in Figure 5.

**Figure 5: The correlations of 21-year running windows between tide gauges (Stockholm-STO and Warnemünde-WAR) and the NAO-index for the winter (dashed line) and summer (solid line) seasons. The value of correlation significance at the 95% level (two-tailed) is ±0.43 for this record length. The correlation coefficients are computed based on de-trended seasonal means between tide gauge records and atmospheric indices.**

Figure 5 shows that the values of the 21-year running correlations are mostly non-significant in summertime. In wintertime, correlations between Stockholm and the NAO-index are weak or not significant until 1965. The Warnemünde station does not seem to be strongly connected to the NAO-index until 1998 in wintertime.

Given that the running correlations are not stable in time for both seasons over the period 1900-2013, it is hypothesised in this sub-section that the temporal instability of the link between stations and the NAO could indicate the existence of another
atmospheric circulation pattern that both strongly affects sea-level variation and that differs from the traditional NAO pattern. Therefore, we further analysed the strength of the relation between the NAO and nine tide gauges by considering the periods in which the NAO had the strongest and the weakest connections to the sea-level variations. This analysis is also separately conducted for wintertime and summertime, respectively.

Figure 6 shows the correlation patterns between the NAO and nine tide gauges in 21-year windows in which these
correlations were highest (1976-1996) and lowest (1950-1970) over the period 1900-2013 in wintertime.

**Figure 6: The correlation maps between de-trended tide gauges and the de-trended NAO-index for the minimum (1950-1970) and the maximum (1976-1996) correlation periods in wintertime.**

For the summer season, the 21-year period with the weakest connection between the NAO-index and tide gauges is 1924-1944, whereas the 21-year period with the strongest correlation is 1960-1980. The correlation maps derived from these two
periods are shown in Figure 7.

**Figure 7: The correlation maps between de-trended tide gauges and the de-trended NAO-index for the 1924-1944 and 1960-1980 periods when minimum and maximum correlations occur in summertime, respectively.**

Figure 7 also illustrates that the difference in sea level-NAO correlation between both periods is spatially rather homogenous, i.e. correlations tend to be low in 1924-1944 for all tide gauges.

**4.4 Definition of the BANOS-index**

Given these results, the question arises as to whether the NAO is the atmospheric pattern most closely related to sea-level variability in the study area. Addressing that question, we decided to assess the correlation patterns between tide gauge records and SLP field based on the 21-year running windows for the period 1900-2013.

We carried out several analyses taking different periods into account.

First, we considered the period 1976-1996 (1950-1970), when the correlation between the NAO-index and the sea-level variability was maximum (minimum) in wintertime. The correlation pattern between SLP field and the Stockholm sea-level over this period is illustrated at the right (left) panel of Figure 8.

**Figure 8: The correlation maps for the periods when the correlation between the NAO-index and sea-level variability were minimum (left) and maximum (right) in wintertime. The periods cover the years 1950-1970 (left) and 1976-1996 (right). The time series are de-trended prior to correlation calculations. The SLP grid cells that are selected to construct the BANOS-index are marked with squares.**

Although, the correlation pattern in Figure 8-right exhibits very similar pattern to the traditional NAO pattern,

Figure 8-left displays an atmospheric pattern that differs from the typical NAO pattern. In particular, the implied direction and strength of geostrophic winds show discrepancies compared to the NAO pattern.

Since this pattern looks different from the typical NAO pattern, we constructed a new index that should reflect the SLP gradient along a different direction from what the NAO implies. First, two grid-cells with the maximum and minimum correlations to the Stockholm tide-gauge were identified. Second, from those grid-cells, the differences of the normalized SLP values were computed to construct a new circulation index that is in the following denoted as the BANOS-index. The geographical points of the SLP grids are (5° W, 45° N) and (20° E, 70° N) for wintertime. These two grid-cells are held fixed, and define the BANOS-index for the whole period 1900-2013.

The same steps are followed in order to construct the BANOS-index for the summer season. In this case, the correlation patterns with the SLP fields are calculated for the period 1924-1944 and 1960-1980 when the link between the NAO-index and the sea-level variations are weakest and strongest, respectively. The associated correlation maps are shown below in Figure 9.

**Figure 9: The correlation patterns for the periods when the correlation between the NAO-index and sea-level variability were minimum (left) and maximum (right) in summertime. The periods cover the years 1924-1944 (left) and 1960-1980 (right). The time series are de-trended prior to correlation calculations. The SLP grid cells that are selected to construct the BANOS-index are marked with squares.**

The correlation pattern obtained when the relation between stations and the NAO was weak and is different from the NAO pattern in summer. Based on the corresponding pattern, we also constructed the BANOS-index for summertime. For this construction, the geographical points of the SLP grid-cells were (30° E, 45° N) and (20° E, 60° N).

## 4.5 Comparison between the BANOS-index and the NAO-index

After constructing the BANOS-index, we compared it to the traditional NAO-index. For this comparison, we first assessed the direct relation between two indices for the winter and summer seasons for the entire period 1900-2013. The correlations between these two indices are 0.68 and -0.12 for winter and summer seasons, respectively. This indicates that the winter BANOS-index shares some similarities with the NAO-index in winter, but it is quite different from the NAO index in summertime. The time series of the indices are represented in Figure 10.

**Figure 10: The time series of the BANOS-index and the NAO-index together with 21-year running means for the winter (top) and summer (bottom) seasons over the period 1900-2013.**

In a next step, we investigated the link of the indices to the Stockholm and Warnemünde tide gauges from 1900 to 2013 and their stability over time. For this investigation, we used 21-year gliding windows in order to examine the connection between these stations and the two indices, also the indices are compared to each other. The time evolutions of the gliding correlations are displayed in Figure 11.

**Figure 11: The correlations between selected tide-gauges and the BANOS and NAO indices based on the de-trended 21-year running windows for the winter (top) and summer (bottom) seasons. Time series are de-trended in every 21-year period prior to the correlation computations. The value of correlation significance at the 95% level (two-tailed) is ±0.43 for this record length.**

In wintertime, the BANOS-index is more strongly correlated with the Stockholm and Warnemünde stations than the NAO-index throughout the whole period. The correlations of 21-year moving windows between Stockholm and the BANOS-index

are significant over the whole period 1900-2013 in wintertime. The highest correlation (r>0.94) between Stockholm and BANOS-index occurs during the satellite era (1993-2013). Considering the correlation between the BANOS-index and Warnemünde, it is shown that the strength of relation is significant with the exception of the period 1927-1937. The strongest correlation between the BANOS-index and Warnemünde (r~0.88) is obtained in the period 1991-2011.

This behaviour is in contrast with the gliding correlations between these two stations and the NAO. The gliding correlations

between the NAO-index and two stations in winter start to increase from 1960 onwards. Notably, the gliding correlations between the stations and the NAO-index resulted in weak or non-significant relations (r<0.43) most of the time over the period 1900-2013 in wintertime. In addition, the Cuxhaven station behaves similarly to Stockholm regarding its moving correlation to the both atmospheric indices.

In summertime, the running correlations between the BANOS-index and the sea-level variability in Stockholm indicate a

stronger link than the one between the NAO-index and Stockholm over the whole period 1900-2013. The maximum correlation value between the BANOS-index and Stockholm (r=0.89) is calculated in the period 1928-1948. It should also be noted that the connection between Cuxhaven and BANOS-index is most of the time significantly correlated, but that was not the case for the NAO-index. The variability of the sea-level in Warnemünde is also strongly connected to the BANOS-index. The highest correlation between the BANOS-index and Warnemünde sea level (r>0.76) occurs during the period 1985-2005.

As in the case of wintertime, this strength of the relation between associated stations and the BANOS-index is in strong contrast with the link between the two stations and the NAO-index. Also, a negative trend is detected in the time series of running correlation between the NAO-index and the two stations starting from year 1970 onwards in summertime. Overall, no significant correlation between the NAO and stations is estimated in summertime.

There are some other local atmospheric circulation patterns than the newly constructed SLP BANOS pattern. It can be

argued that the patterns such as the Scandinavia, the Polar/Eurasia and the East Atlantic are close to the region, thus, may modify sea level in the Baltic Sea and the North Sea. To depict the the covariability between the indices of those patterns and sea level variability in the Baltic Sea and the North Sea region, we analysed the strength of the relation between the representative tide gauges (Stockholm, Warnemünde and Cuxhaven) and three teleconnection pattern indices in wintertime and summertime for the period 1950-2013. Please see the table showing the correlation computations below.

**Table 2: The correlations between selected tide gauges and the indices of teleconnection patterns for the period 1950-2013. Time series are detrended prior to the correlation computations and the 95% significance level is ±0.24 for this record length.**

The correlation table indicates that the sea level variation in the Baltic Sea and the North Sea tide gauge stations mostly weakly covary with those teleconnection patterns. In wintertime, stations have mostly non-significant correlation values. Only the Scandinavian teleconnection pattern implies significant relation with the Stockholm and Cuxhaven tide gauges in wintertime. In summertime, the correlations are also not quite strong. The largest correlation value measured between Scandinavian pattern and the Stockholm and Cuxhaven stations is -0.47.

## 4.6 Influence of the BANOS pattern on off-shore sea-level variability

After assessing the time evolution of the link between the BANOS-index and the Stockholm and Warnemünde stations, we quantify the off-shore sea-level variability connected to the BANOS-index. The relation between satellite altimetry SLA grids from the AVISO product and the BANOS-index was analysed over the period 1993-2013. This analysis is carried out using correlation of de-trended time series for the winter and summer seasons. The correlation patterns for both seasons are shown in Figure 12.

**Figure 12: The correlation patterns between the de-trended SLA grids and the de-trended BANOS-index for the winter (top) and summer (bottom) seasons over the period 1993-2013. The 95% significance level is ±0.43. The areas indicating significant correlations are delineated with contour lines.**

In general, the connection between the BANOS-index and the off-shore sea-level variability is found to be strong over most of the study area for both winter and summer seasons, confirming the strong relationship between the BANOS-index and two tide gauges. In wintertime, the correlation patterns of the off-shore sea-level variability with the BANOS-index show that the strongest relation is located in the Baltic Proper and Gulf of Riga, with the value of r~0.95. The correlation pattern decreases (r~0.83) over the transition zone between the Baltic Sea and the North Sea. In the North Sea, the connection is strongest (r~0.93) in the German Bight area. The weakest relation occurs in the Skagerrak area, where correlation decreases to 0.28 (not significant).

Comparing the correlation maps showing the relation of SLA grids to both the BANOS-index and the NAO-index (Figures 4 and 12), it is seen that the BANOS-index is more closely connected to the off-shore sea-level variability than the NAO-index over the Baltic Sea and North Sea region.

In summertime, coastal and off-shore sea-level variations seem to be well connected to the BANOS-index, in contrast to the non-existent relation between the NAO-index and sea-level variability in the Baltic Sea and North Sea region, there is a spatially continuously increasing correlation from Skagerrak to Arkona Basin, ranging from -0.01 to 0.75. The maximum correlation value is detected in the central Baltic and in the eastern Baltic, reaching to 0.89. This means that the BANOS-related atmospheric circulation pattern explains up to 79% of the sea-level variance in the Baltic Sea in summertime. Referring to Figure 1, the associated sub-regions are the Baltic Proper, the Bothnian Sea and the Gulf of Finland.

Considering the North Sea, the link between the BANOS-index and sea-level variability is found to be strongest in the eastern part of the North Sea, with a maximum correlation of up to r~0.63. Apart from this region, no substantial correlation is found in the North Sea in the summer season.

### 4.7 Sensitivity of satellite sea-level variations to the BANOS-index

Given not only the relatively stable correlation in time between the BANOS-index and coastal sea-level variations compared to the NAO, but also the high correlation between the BANOS-index and the satellite altimetry SLAs over the entire Baltic Sea and a part of the North Sea, we further quantify the linear response of off-shore sea-level variability to the BANOS-index. For this purpose, the sensitivity of the sea-level to the BANOS-index is estimated from the linear regression parameter of the linear regression where the BANOS index is the predictor and the sea-level at each grid-cell is the predictand. Before

estimating the sensitivity of sea-level, we linearly de-trended the time series of the associated predictand and predictor, since the sea-level records contain the trend caused by global sea-level rise and crust movements which are not related to the variability of the atmospheric circulation. The corresponding sensitivity values of the SLA grids are represented in Figure 13.

**Figure 13: The sensitivity values of the SLAs to the BANOS-index for the winter (left) and summer (right) seasons over the period**
**1993-2013. Note the different intervals on the colour scales.**

In wintertime, the largest sensitivity of the SLA to the BANOS-index appears in the eastern part of Gulf of Finland, with values reaching to 92 mm per one unit (mm u$^{-1}$) (r>0.90) change in the BANOS-index. Another large sensitivity is detected in the northeast Bothnian Bay with the value of 81 mm u$^{-1}$ (r=0.90). The sensitivity values are ranging from 77 mm u$^{-1}$ to 80 mm u$^{-1}$ (r=0.95) in the Gulf of Riga. Considering the North Sea region, the maximum sea-level sensitivity is calculated for

the German Bight; 60 mm u$^{-1}$ (r=0.93).

In summertime, the Gulf of Riga is found to be the most sensitive area with value of 31 mm u$^{-1}$ (r=0.89). The eastern Gulf of Finland has a sensitivity reaching up to 29 mm u$^{-1}$ (r=0.86). In the German Bight, we detected a value of 14 mm u$^{-1}$ (r>0.48), where the sensitivity of the North Sea to the BANOS-index is the strongest during summer. Notably, Skagerrak was the least sensitive region to the BANOS-index in the winter (5 mm u$^{-1}$) and summer (-1 mm u$^{-1}$) seasons over the study area.

Although the sensitivity values would change depending on the exact definition of the BANOS-index (for instance, whether or not the definition involved standardization to unit variance), the sensitivity parameters  describe the relative sensitivity of associated sea-level in different locations with respect to other areas in the Baltic Sea and North Sea region.

### 4.8 Possible physical factors contributing to the sea-level variability

The aim of this sub-section is to investigate the physical mechanism(s) that may explain the link between the BANOS-index
and sea-level in the Baltic Sea and North Sea region on the interannual time scale. For this purpose, we estimate the portion of the variances of sea-level that is statistically explained by different physical factors. We examined several plausible

candidates, as explained in the following.

It is known that the SLP can substantially impact the sea-level variations by a direct response due to the inverse barometer effect (IBE). Also, the BANOS patterns suggest that geostrophic winds could influence the sea-level variations over the Baltic Sea and the North Sea. Net energy flux variations linked to the BANOS index can also affect sea-level variation. The possible BANOS-attributed heat flux storage can cause thermosteric rise (volumetric) in the sea level. Additionally, the BANOS pattern may carry information about other mechanisms (i.e. variation in precipitation, evaporation) which may explain the contribution of the freshwater flux to the sea-level variability over the Baltic Sea and North Sea region. Accordingly, freshwater flux contribution would indicate the significant effect of BANOS related net precipitation mass variation in the sea level. We also explore the effect of the wind on the off-shore water movement due to the Ekman transport.

### 4.8.1 The contribution of the inverse barometer effect (IBE)

One of centres of action of the BANOS patterns lies over the Baltic Sea, and therefore the influence of the IBE on sea-level variations seems to be a plausible factor which can explain the physical mechanism between the BANOS-index and sea-level on interannual time scales. By the investigation of IBE contribution, we assumed the presence of an infinite ocean without topographic limitation and complete equilibrium in the Baltic Sea and North Sea regions to the overlaying air pressure. This assumption implies that water is free to move responding to air-pressure gradients to reach hydrostatic equilibrium. This is a simplifying assumption that nevertheless allows to estimate an order of magnitude of the contribution of the BANOS-related IBE to sea-level variations. The BANOS pattern shows that lower air pressure over the Baltic Sea region and higher pressure around the Gulf of Biscay (over the area between Labrador Sea and Denmark Strait) in wintertime (summertime) are connected to higher sea-level in the Baltic Sea. Therefore, we first selected the geographical points by considering the BANOS patterns in order to estimate the impact of the pressure differences on the sea-level. For wintertime, the coordinates were (5° W, 50° N) - (20° E, 60° N) and for summertime they were (30° W, 65° N) - (20° E, 60° N). The choice of these points was dictated by the centres of action of the BANOS pattern constrained by the availability of sea-level data. To test the possible influence of the IBE on sea-level, we initially took the differences of sea-level and the SLP fields over the given geographical points for each season. Then, we implemented a linear regression over the period 1993-2013 where differences of sea-level variations were the predictand and differences of SLP fields were the predictor. The sea level sensitivity values are 18.1 mm and of 7.0 mm per 1 hPa change in the described SLP field differences for wintertime and for summertime, respectively.

In the next step, we investigated the sensitivity of the SLP differences per one unit change in the BANOS-index for the period 1900-2013. The linear regression between the BANOS-index (predictor) and the SLP differences (predictand) results in a sensitivity of 3.44 hPa and 1.39 hPa per one unit change in the BANOS-index in wintertime and in summertime, respectively. These results indicate a large contribution of the air pressure differences on the interannual variation in the sea-level over the selected points, estimated as of 62.2 mm u$^{-1}$ (3.44*18.09) in wintertime and of 9.8 mm u$^{-1}$ (1.39*7.02) in

summertime per one unit change in the BANOS-index.

This means that assuming a complete equilibrium of sea-level to the SLP differences over this region, one unit increase in the BANOS-index would cause up to 62.2 mm (9.8 mm) rise in the sea-level during wintertime (summertime) due to the IBE in the Baltic Sea and the North Sea. More importantly, the correlation analysis between BANOS-index and differences in SLP fields suggests that 88% (34%) in wintertime (summertime) sea-level variance linked to the BANOS pattern can be accounted for by the IBE.

### 4.8.2 The contribution of net surface energy flux

Net surface energy flux (NEF), the total energy transfer through the Earth surface, is composed of radiative and turbulent fluxes. The radiative fluxes are composed of shortwave (SW) solar radiation reaching to the earth surface and longwave (LW) emitted energy from the Earth surface. The turbulent fluxes are sensible heat (SH) and latent heat (LH) fluxes.

The NEF is the difference of associated fluxes between the energy absorbed by the Earth and the energy emitted from the surface of the Earth. Here, we simply re-expressed this relation by using following definition (Eq. 1).

$$NEF+^{\downarrow} = SW_{net} - (LW_{net}+SH_{net}+LH_{net}), \tag{1}$$

NEF is the net energy flux, defined here positive downward. $SW_{net}$ is the net downward shortwave radiation, $LW_{net}$ is the net upward longwave radiation, $SH_{net}$ is net upward sensible heat flux and $LH_{net}$ is the net upward latent heat flux. After computing the NEF for the winter and summer seasons over the period 1949 to 2013, we calculated the correlation values between individual heat flux in each grid-cell of the NCEP/NCAR meteorological analysis and the BANOS-index. The correlation patterns are illustrated in Figure 14.

**Figure 14: The correlation patterns based on de-trended time series between the NEF(+↓) and the BANOS-index in wintertime (left) and summertime (right) over the period 1949-2013. For this record length, the 95% significance level is ±0.24. The areas indicating significant correlations are delineated with contour lines.**

The correlation pattern indicates a strong connection between the NEF and the BANOS-index over the Baltic Sea and North Sea region in wintertime. This is consistent with the correlation pattern between the SLA grids and the BANOS-index for the winter season. However, in summertime the correlation pattern has negative values over the study area. This implies that the heat fluxes linked to the BANOS-index are not responsible for the sea-level variations over the Baltic Sea and North Sea region during summertime, although they can oppose the sea-level variations caused by other factors. Therefore, we only considered wintertime in order to estimate the contribution of the NEF variations to the connection between the BANOS-index and sea-level. For this computation, we took the spatial average of the NEF values considering the geographical window between the (0° - 30° E) longitudes and the (50° N - 72° N) latitudes.

The sensitivity value of the NEF to the BANOS-index is estimated as 3.28 (W m$^{-2}$u$^{-1}$) (r=0.59) in wintertime. This represents an average increase in energy (absorbed-minus-emitted) of 2.5e7 (J m$^{-2}$yr$^{-1}$) per one unit change of the BANOS-index over one winter. This energy storage can be translated into an approximate estimation of sea level rise, assuming that the specific

heat and thermal expansion of sea-water does not depend on water temperature, salinity or water pressure. Based on this rather strong assumption, we estimated the strength of the relation between sea-level variability linked to the BANOS-index and spatially averaged NEF. This analysis suggests that 35% of the BANOS-related sea-level variability in the Baltic Sea and North Sea region can be explained by the NEF contribution in wintertime. This estimation shows the order of magnitude

of the net energy flux contribution to explain the linkage between the BANOS-index and sea level variability. Accordingly, the amount of relative thermal expansion of the water per one unit change in the BANOS-index could be computed as well. However, this computation would differ depending on the assumed average value of temperature and pressure through the water column. An estimation shows that 1 unit increase in the BANOS-index can cause 1 mm sea level rise due to the contribution of net energy flux. This estimation is independent of water depth under the assumption that the thermal

expansion coefficients do not depend on temperature or pressure.

### 4.8.3 The contribution of freshwater flux

The BANOS patterns also suggest possible effects of freshwater flux on sea-level variability. The freshwater flux has two different components. One is precipitation (P) showing complex space-time pattern over the study area. The other is evaporation (E), associated with the net latent heat loss of the surface. The difference of these two factors (P-E) is defined as

the freshwater flux.

Here, we used the latent heat flux from the NCEP/NCAR meteorological reanalysis in order to approximately compute the evaporation rate. The surface evaporation rate (E: mm s$^{-1}$) can be approximately computed by using Eq. 2.

$$E = \frac{Q_{lat}}{L_e \rho_W},$$
(2)

where $Q_{lat}$ is the latent heat flux (W m$^{-2}$), $L_e$ is the latent heat of water vaporization (2257 kJ kg$^{-1}$) and $\rho_W$ is the freshwater

density (1000 kg m$^{-3}$).

We then display the correlation patterns between the BANOS-index and the freshwater flux and between the BANOS-index and precipitation and evaporation separately in order to examine the possible impact of the freshwater flux on the sea-level variability linked to the BANOS pattern. The correlation patterns of the BANOS-index with precipitation, evaporation and freshwater flux are shown in Figure 15.

**Figure 15: The correlation patterns of precipitation (top), evaporation (middle) and freshwater flux (bottom) to the BANOS-index in wintertime (left) and summertime (right) over the period 1949-2013. Time series are de-trended prior to the correlation computation. The 95% significance level is ±0.24. The areas indicating significant correlations are delineated with contour lines.**

In summertime, the correlation patterns between the BANOS-index and precipitation and between the BANOS-index and freshwater flux display similar results, with the exception of the western part of the North Sea. In addition, correlation

patterns between evaporation and the BANOS-index indicate that evaporation patterns appear to be partly connected to the BANOS-index over the Baltic Sea and North Sea region in the summer season. The results of these correlation patterns suggest that the effect of evaporation is opposite to the possible contribution of precipitation to the BANOS-driven sea-level

variability in summertime. Thus, we further focused on the strength of the relation between freshwater flux (P-E) and BANOS-related sea-level variability in this region.

It should be noted that P-E does not cause sea-level to vary directly on the geographical points where they occur, but they have to be considered over the whole Baltic Sea catchment basin.

The corresponding correlation analysis between the freshwater flux averaged over the Baltic catchment basin and the BANOS-index indicates that 27% variance of the sea-level variability that was attributed to the BANOS-index in the Baltic Sea can be explained by freshwater flux variations. However, contribution of the BANOS-related freshwater flux to sea level variation is very small for the North Sea.

In wintertime, the correlation patterns between the BANOS-index and the P-E vary from strongly negative in the Baltic
Proper to strongly positive into the eastern side - covering Gulf of Riga and Gulf of Finland - and the western side of the Baltic Sea centre, namely including the area Arkona basin, Danish straits, Kattegat and Skagerrak. However, considering the basin wide connection between the P-E and the BANOS-driven sea-level variability in the Baltic Sea and the North Sea, the effects of those factors on the sea-level variability seem to be negligible in wintertime.

Our results suggest that freshwater flux considerably contributes to the BANOS-driven sea-level variability only in the Baltic Sea region and only in summertime. In the light of this finding, we further investigated the relation between freshwater flux and the BANOS-index in order to see the spatial evolution of the freshwater flux based on the BANOS pattern of atmospheric circulation. Therefore, we estimate the sensitivity values of freshwater flux at reanalysis grid-cell per one unit change in the BANOS-index. The associated sensitivity patterns of the freshwater flux grids are shown in Figure 16.

**Figure 16: The sensitivity patterns of the freshwater flux grids per unit change in the BANOS-index in wintertime (left) and**
**summertime (right).**

In the right panel of Figure 16, the summer sensitivity pattern between freshwater flux and the BANOS-index shows that freshwater flux is relatively more sensitive over large parts of the Baltic Sea drainage basin with respect to the rest of whole study area. Figure 16 also illustrates that the sensitivity of freshwater flux is largest over the eastern part of Baltic Sea drainage basin to one unit change in the BANOS-index throughout the summer season. An estimation considering the basin wide average of freshwater flux and the BANOS-index suggests that the sensitivity value of sea-level would reach 10 mm per one unit change in the BANOS-index due to the freshwater flux effect in the Baltic Sea region over the summer season.

### 4.8.4 The contribution of geostrophic wind forcing

Another plausible factor contributing to the linkage between the BANOS-index and sea-level variability in the Baltic Sea and North Sea region would be the BANOS-related wind. In this sub-section, we aim at explaining the transport of water due to the wind forcing related to the BANOS pattern.

In wintertime, the geostrophic wind flow linked to the BANOS pattern (Figure 8-left) is slightly different to that indicated by the NAO pattern (Figure 8-right). This may result in different responses to wind forcing in the Baltic Sea and North Sea region. Considering the area including north-eastern North Sea, Skagerrak and Kattegat (Figure 1) that slight modification in

the wind direction suggests that the associated wind forcing cannot transport surface water from the North Sea into the Baltic Sea. This result is in contrast to the NAO-related wind forcing.

In summertime, the BANOS-related (Figure 9-left) wind seems to generate similar direction of the water transport as the NAO pattern (Figure 9-right) does over the transition zone. This indicates that the wind forcing mechanism in summertime can be similar to the case of the NAO.

Figure 17 shows the estimated Ekman transport caused by the wind related to the BANOS pattern assuming that the Ekman layer is not interrupted by bathymetry.

**Figure 17: The vectors represent the direction of the Ekman transport caused by the SLP BANOS patterns, assuming a geostrophic wind approximation and a complete ocean Ekman layer. The lengths of the vectors are proportional to the magnitude of the surface wind, and directed 90 degrees to the right of the surface wind. Upper (lower) panel shows the winter (summer) season Ekman transport vectors.**

The vectors shown in Figure 17 represent the estimated Ekman transports in the Baltic Sea and North Sea basins. For instance, concerning only the North Sea basin, if bathymetry would not interrupt the Ekman spiral, Ekman transport generated by the BANOS-related north-easterly winds, would be directed in south-westward direction towards the German, Dutch and UK coastlines, in fact depleting sea-level in the Baltic Sea. For the wind forcing, to be relevant in explaining the link between the BANOS pattern and Baltic Sea level, the assumption of a complete Ekman spiral needs to be dropped, so that the transport could flow more parallel to the surface wind. Even in that case, an atmospheric circulation pattern more efficiently driving higher Baltic Sea level should have presented a geostrophic wind anomalies more oriented in the west-east direction.

### 4.8.5 The contribution of open ocean sea-level variability

Coastal sea-level can also be affected by off-shore sea-level variations in the open ocean. For instance, under spatially uniform warming, sea-level at the coast will rise more than corresponding to its smaller thermal expansion of its shallower water column due to the partial transfer of water mass from the open ocean. Since the deep ocean water is expected to have stronger expansion than the shallow water due to its deeper water column. (see e.g. Grinsted et al., 2015). In the context of climate change scenarios, this effect have been estimated for the Baltic Sea as proximately an additional 10% sea-level rise at the coast. In our case, the estimation of this effect would entail the calculation of the effect of the BANOS pattern to temperature changes in the open ocean (North Atlantic) in addition to the expansion of the water column directly in the Baltic Sea. The observations needed for this calculation are not available for the whole 20[th] century, and ocean reanalysis cover only the last few decades, thus this calculation could only refer to a fraction of the analysis period. However, Figure 13 already indicates that this effect cannot be large for the interannual variability of sea-level in the Baltic Sea. The pattern showing the sensitivity of sea-level to variations of the BANOS pattern is constrained to the Baltic Sea and shows very small, even negative sea-level anomalies in the open ocean. Therefore, since sea-level in the open ocean is not higher for positive phases of the BANOS pattern, this transfer of water mass to the coastal areas and to the Baltic Sea cannot take place. This interpretation is also supported by Figure 14, which shows the correlation pattern between the BANOS index and net

surface heat flux. The heat flux anomalies linked to the BANOS index are negative in large portion of the North Atlantic, albeit they are positive at the south of the British Channel.

**5 Discussion and conclusions**

We have identified an atmospheric circulation pattern that represents a more stable and stronger statistical connection to sea-level variations in the Baltic Sea than the better known NAO pattern. These results suggest that the BANOS pattern may represent a more effective forcing of sea-level by the atmospheric circulation. However, this result does not exclude the influence of other patterns of atmospheric circulation. Actually, the BANOS and the NAO patterns are statistically not independent and both describe a kind of SLP gradient, but with different orientation. Atmospheric variability that is described by the North Atlantic Oscillation also causes variations in the BANOS index. The relative importance of each mode of the atmospheric circulation for the Baltic Sea level variability may vary for different periods through the 20[th] century as it is clearly seen for the NAO. This influence depends on two factors. One is the link between the circulation pattern and the Stockholm, Cuxhaven and Warnemünde stations, the second factor is the amplitude of variations of the atmospheric index itself. Therefore, it is possible that for some periods in the past or in the future, the NAO may be more strongly correlated to sea-level than the BANOS index if, for instance, its amplitude of interannual variations becomes larger. However, regarding the long-term picture over the whole 20[th] century, our results indicate that influence of the BANOS pattern on the Baltic Sea level is on average stronger and more stable.

Here, it should be noted that the BANOS mode of atmospheric circulation shares some similarities with the atmospheric proxy that is suggested by (Dangendorf et al., 2014). However, the SLP BANOS pattern indicates different atmospheric pattern than SLP pattern that they describe, especially in summertime. This difference also affects the role of the physical factors that explain the linkage between the BANOS-index and sea-level variability. For instance, we interpret that the BANOS mode of atmospheric circulation does not indicate a wind-driven surface water transport from the North Sea to the Baltic Sea over the transition zone in wintertime. In addition, the correlation pattern between the BANOS-index and SLA grids indicates a large scale seesaw effect of BANOS SLP pattern on sea-level variability between the North Atlantic Ocean and the Baltic Sea region. This large-scale effect is also indicated by the correlation maps between SLA grid and tide gauges. As an interpretation of that seesaw picture effect, our study explains that the IBE plays a key role in explaining the linkage between the SLP BANOS pattern and sea-level variability. This effect was not discussed in the previous studies (e.g. Andersson, 2002; Dangendorf et al., 2014).

It is worth mentioning that the BANOS-index can be used as a proxy for the atmosphere-driven sea level variability. It can be then used to a) estimate the atmosphere-driven sea level variability in previous centuries when sea level measurements were not available, b) project the influence of atmospheric circulation on sea level variability and rise under different climate scenarios for the near future.

The main conclusions that can be drawn from this study are summarized as follows:

1) There exists another pattern of atmospheric circulation that is more strongly correlated to the Baltic Sea level variability. This link is also more stable in time and more homogeneous across space than the link between the NAO and sea-level.

2) The statistical analysis provides a ranking of the possible physical mechanisms that may explain the connection between the BANOS pattern and sea level. The main mechanism appears to be the Inverse Barometric effect. Additionally, in wintertime the surface heat flux anomalies also contribute to the link between the BANOS pattern of atmospheric circulation and sea-level, whereas in summer it is the freshwater flux anomalies.

3) The role of wind forcing and Ekman transport is also seasonally dependent. In wintertime, it appears to not to be an important factor, whereas in summertime it may also explain part of the connection between the BANOS pattern and the Baltic Sea level.

## 6 Acknowledgements

This study was supported by the Deutsche Forschungsgemeinschaft (DFG) through the CliSAP excellence cluster. We thank AVISO for providing satellite altimeter data. Most tide gauge data were obtained from the Permanent Service for Mean Sea Level, and some portions of the Stockholm, Travemünde, Wismar, Warnemünde, Sassnitz tide gauge data sets were kindly provided by Martin Ekman (Stockholm) and Andreas Groh from the Technische Universität Dresden. We are also grateful to NCAR, NCEP and CPC centres for the climatic data sets used in this study.

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

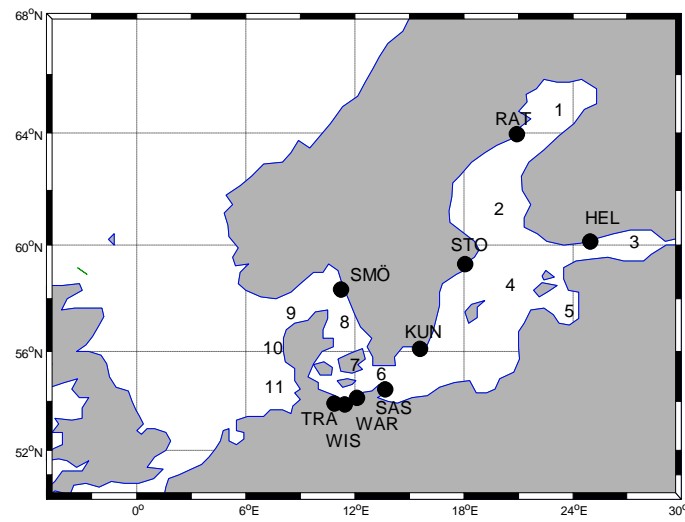

**Figure 1: The Study Area with sub-regions. 1-Bothnian Bay, 2-Bothnian Sea, 3-Gulf of Finland, 4-Baltic Proper, 5-Gulf of Riga, 6-Arkona Basin, 7-Danish Straits, 8-Kattegat, 9-Skagerrak, 10-North-eastern North Sea, 11-German Bight. The study area is also shown together with the tide gauge locations: Helsinki, Sassnitz, Warnemünde, Wismar, Travemünde, Smögen, Kungsholmsfort, Stockholm and Ratan (names are written in an order starting from far east station and follows clockwise rotation).**

**Table 1: The correlations between individual satellite altimetry grids and the tide gauges for the period 1993-2013.**

|  | Winter | Summer |
|---|---|---|
| Helsinki | 0.98 | 0.93 |
| Sassnitz | 0.89 | 0.85 |
| Warnemünde | 0.82 | 0.72 |
| Wismar | 0.89 | 0.78 |
| Travemünde | 0.83 | 0.75 |
| Smögen | 0.88 | 0.76 |
| Kungsholmsfort | 0.95 | 0.91 |
| Stockholm | 0.97 | 0.92 |
| Ratan | 0.95 | 0.91 |

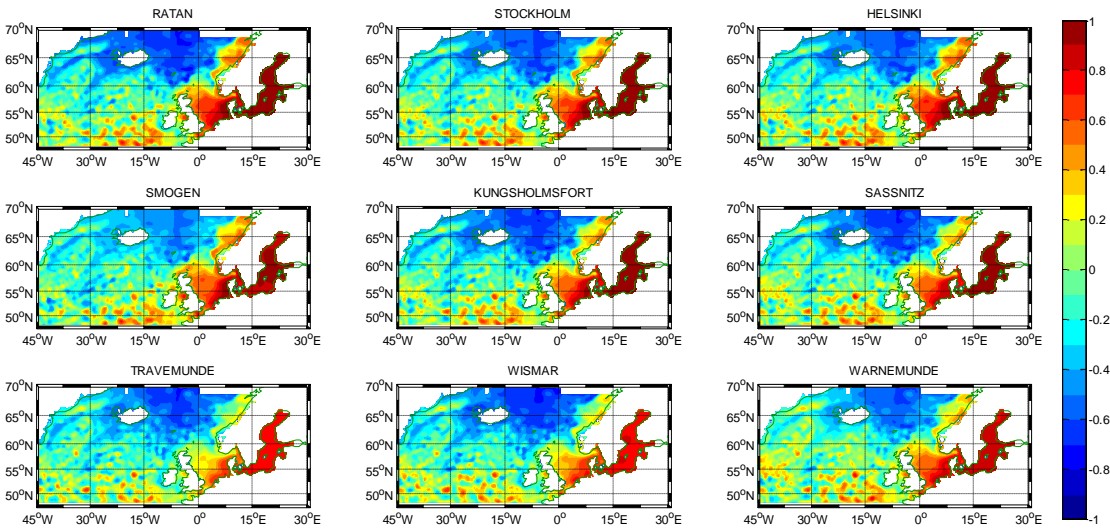

**Figure 2: The correlation patterns between the SLA grids and tide gauge records in wintertime (DJF) for the period 1993-2013. The value of correlation significance is ±0.43 at the 95% confidence level for this record length.**

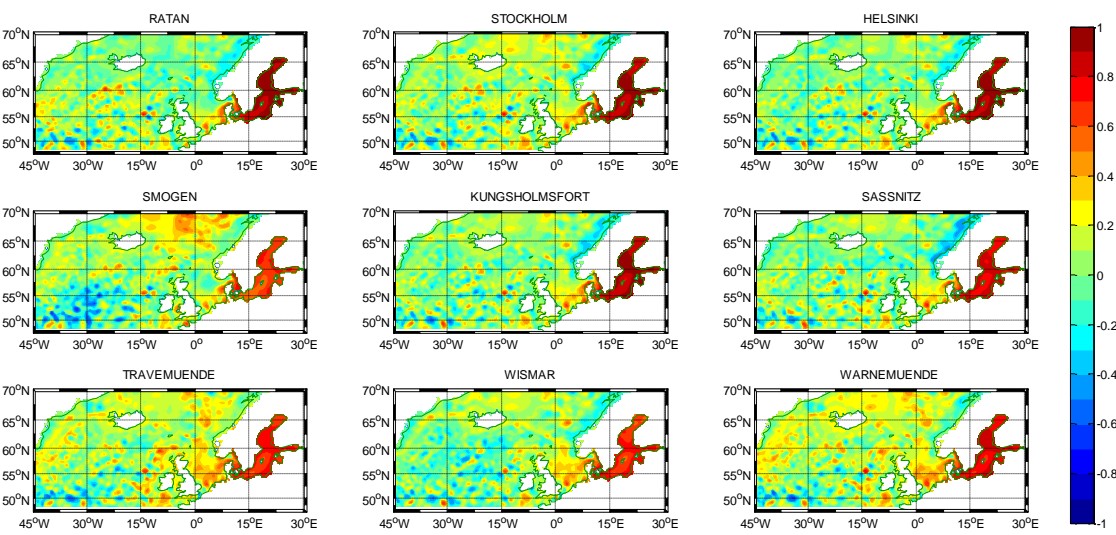

**Figure 3: The correlation patterns between the SLA grids and tide gauge records in summertime (JJA) for the period 1993-2013. The correlation coefficients are computed based on de-trended seasonal means between SLA grids and the tide gauge records. The value of correlation significance is ±0.43 at the 95% confidence level for this record length.**

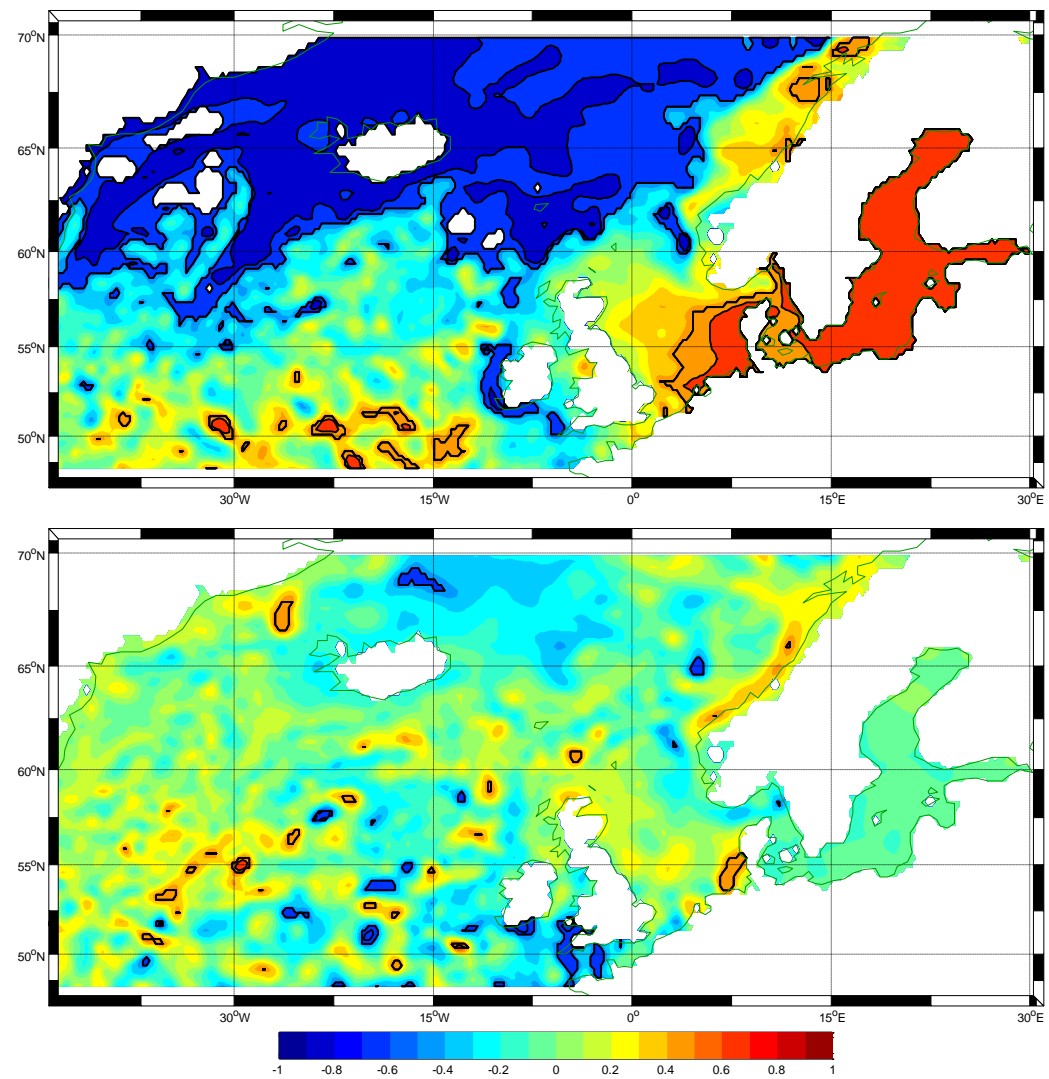

**Figure 4: The correlation maps between the SLA grids and the NAO-index for the winter – means of DJF months - (top) and summer – means of JJA months - (bottom) seasons (1993-2013). The correlation coefficients are computed based on de-trended seasonal means between SLA grids and the NAO-index. The value of correlation significance is ±0.43 at the 95% confidence level for this record length. The areas indicating significant correlations are delineated with contour lines.**

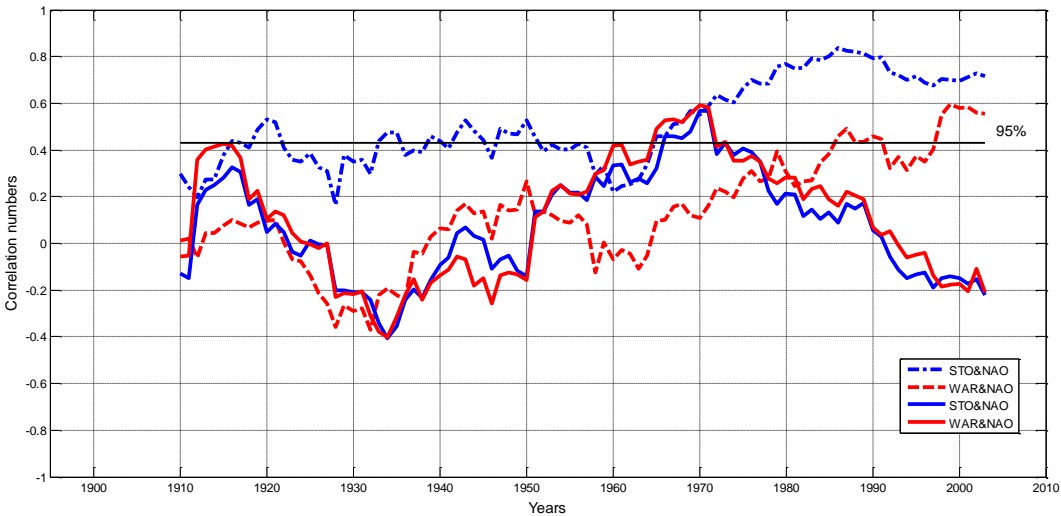

**Figure 5: The correlations of 21-year running windows between tide gauges (Stockholm-STO and Warnemünde-WAR) and the NAO-index for the winter (dashed line) and summer (solid line) seasons. The value of correlation significance at the 95% level (two-tailed) is ±0.43 for this length. The correlation coefficients are computed based on de-trended seasonal means between tide gauge records and atmospheric indices.**

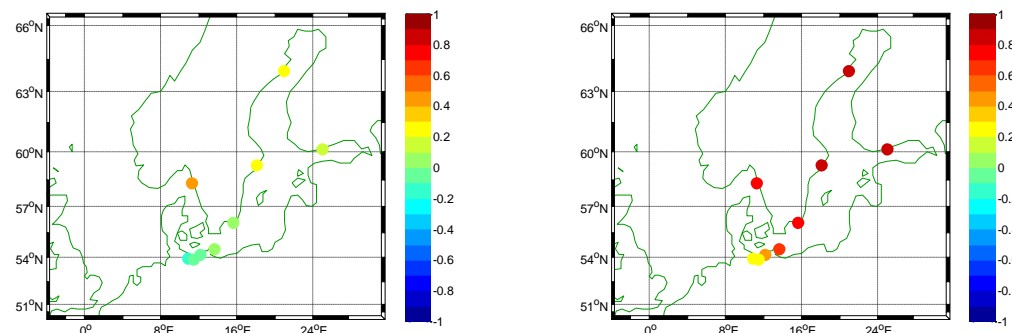

**Figure 6: The correlation maps between de-trended tide gauges and the de-trended NAO-index for the minimum (1950-1970) and the maximum (1976-1996) correlation periods in wintertime.**

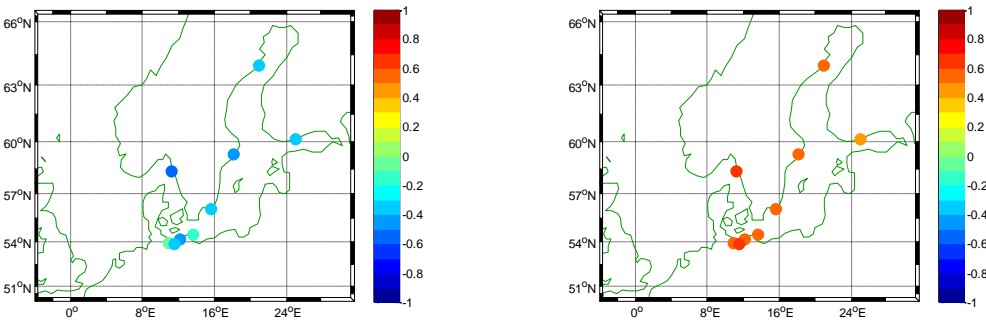

**Figure 7: The correlation maps between de-trended tide gauges and the de-trended NAO-index for the 1924-1944 and 1960-1980 periods when minimum and maximum correlations occur in summertime, respectively.**

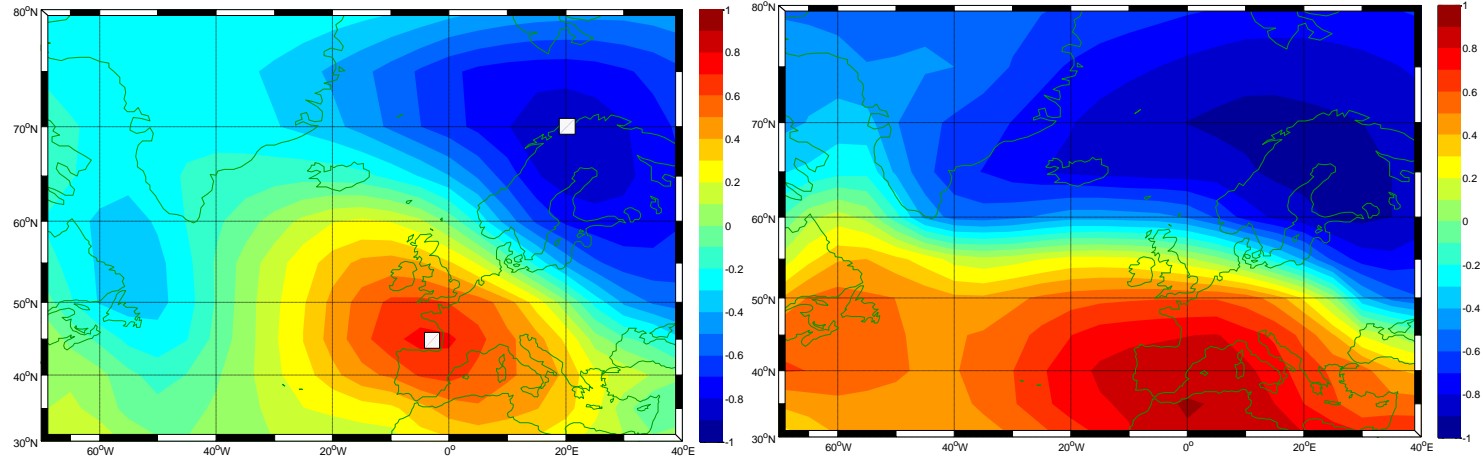

**Figure 8: The correlation maps for the periods when the correlation between the NAO-index and sea-level variability were minimum (left) and maximum (right) in wintertime. The periods cover the years 1950-1970 (left) and 1976-1996 (right). The time series are de-trended prior to correlation calculations. The SLP grid cells that are selected to construct the BANOS-index are marked with square.**

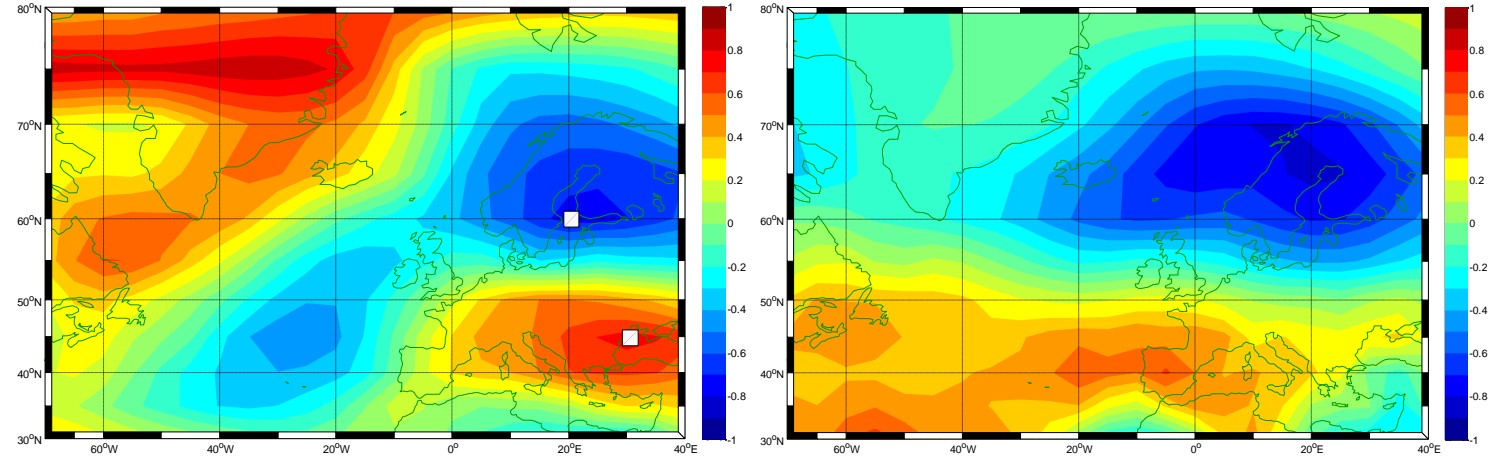

**Figure 9: The correlation patterns for the periods when the correlation between the NAO-index and sea-level variability were minimum (left) and maximum (right) in summertime. The periods cover the years 1924-1944 (left) and 1960-1980 (right). The time series are de-trended prior to correlation calculations. The SLP grid cells that are selected to construct the BANOS-index are marked with square.**

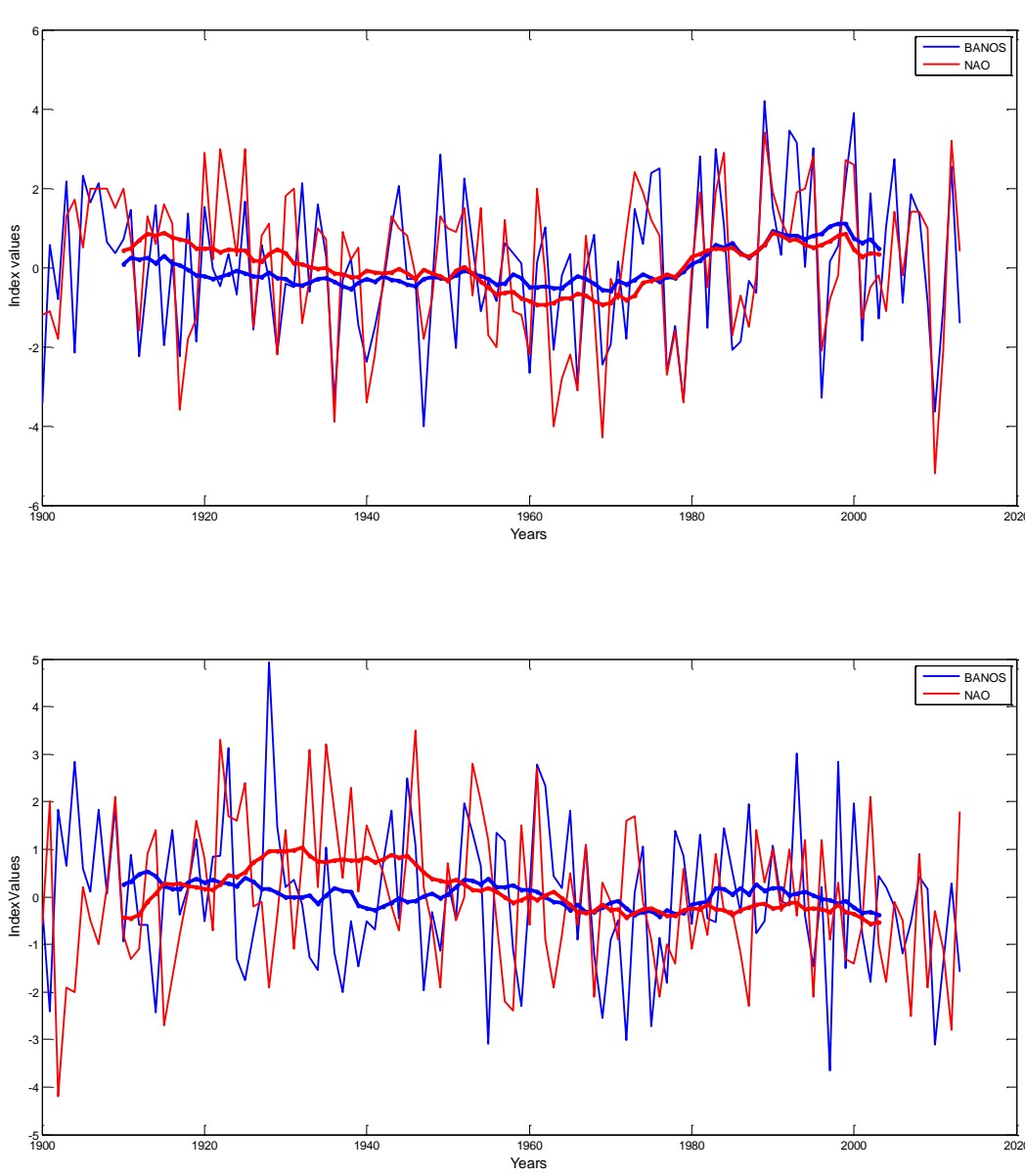

**Figure 10: The time series of the BANOS-index and the NAO-index together with 21-year running means for the winter (top) and summer (bottom) seasons over the period 1900-2013.**

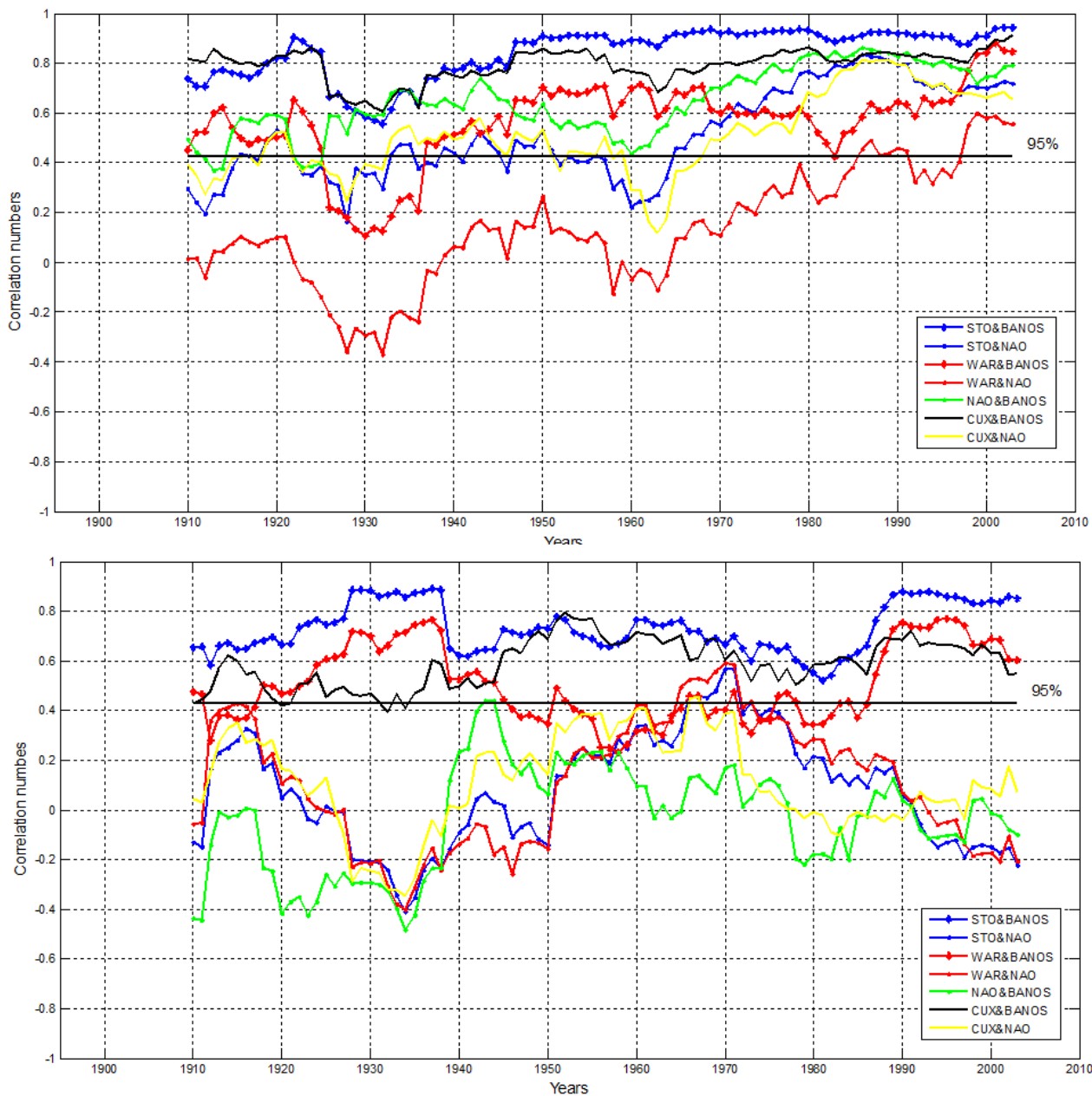

**Figure 11: The correlations between selected tide-gauges and the BANOS and NAO indices based on the de-trended 21-year running windows for the winter (top) and summer (bottom) seasons. Time series are de-trended in every 21-year period prior to the correlation computations. The value of correlation significance at the 95% level (two-tailed) is ±0.43 for this record length.**

**Table 2: The correlations between selected tide gauges and the indices of teleconnection patterns for the period 1950-2013. Time series are de-trended prior to the correlation computations and the 95% significance level is ±0.24 for this record length.**

| Winter/Summer | Scandinavia | Polar Eurasian | East Atlantic |
|---|---|---|---|
| Stockholm | -0.67/-0.47 | -0.03/-0.41 | -0.18/0.28 |
| Warnemünde | 0.11/-0.27 | -0.21/-0.25 | 0.04/0.27 |
| Cuxhaven | -0.60/-0.47 | -0.10/-0.34 | -0.06/0.45 |

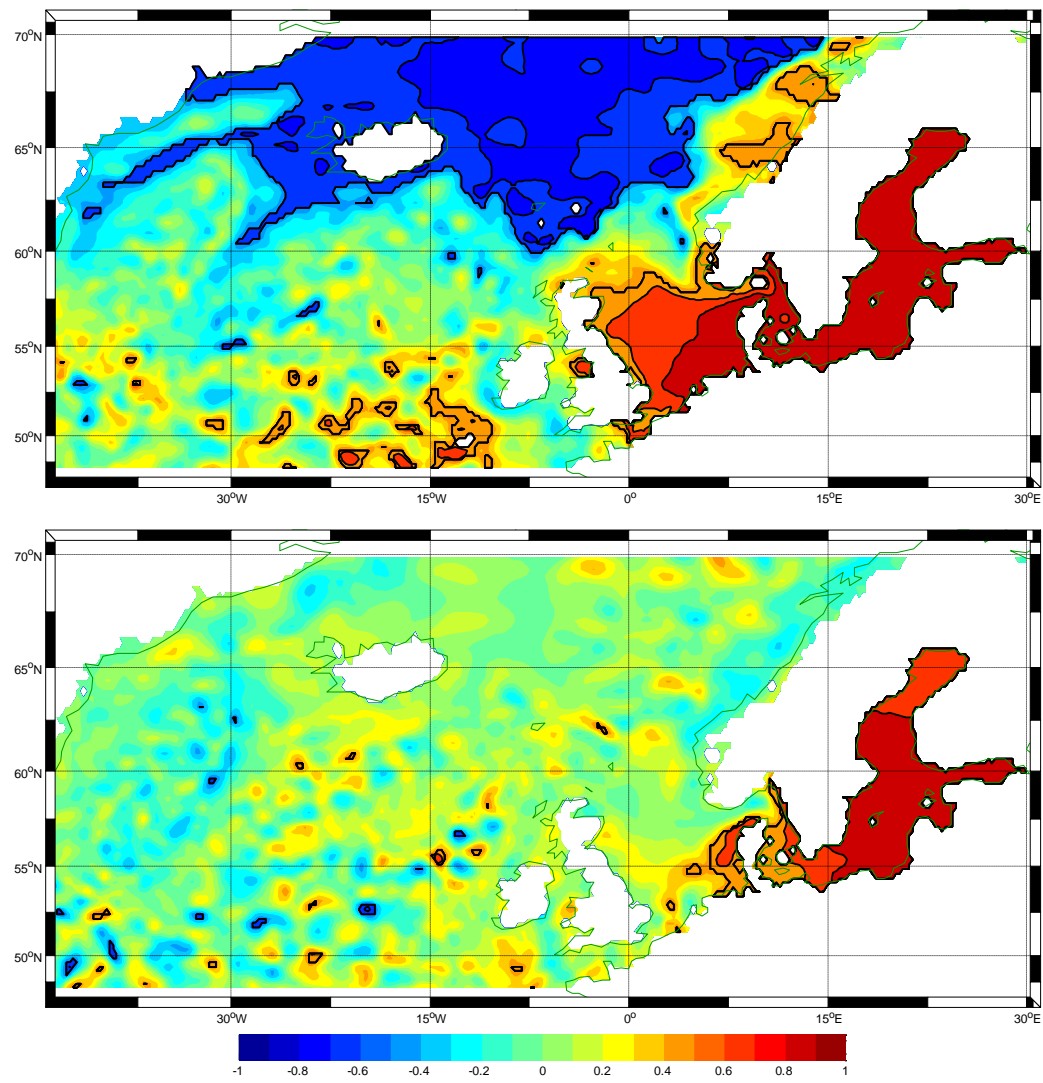

**Figure 12: The correlation patterns between the de-trended SLA grids and the de-trended BANOS-index for the winter (top) and summer (bottom) seasons over the period 1993-2013. The 95% significance level is ±0.43. The areas indicating significant correlations are delineated with contour lines.**

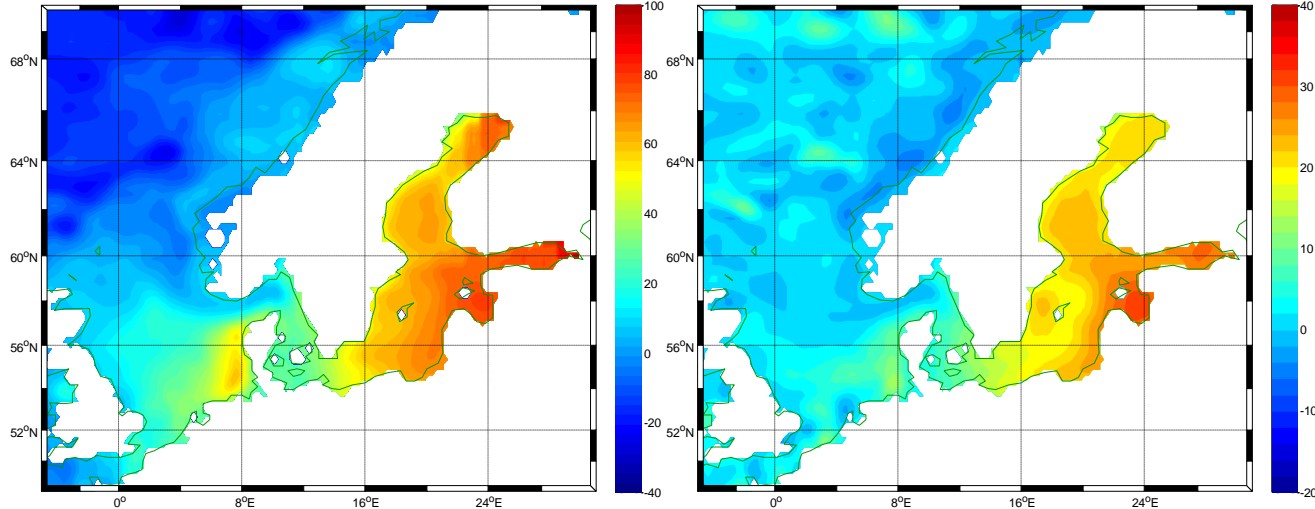

**Figure 13: The sensitivity values of the SLAs to the BANOS-index for the winter (left) and summer (right) seasons over the period 1993-2013. Note the different intervals on the colour scales.**

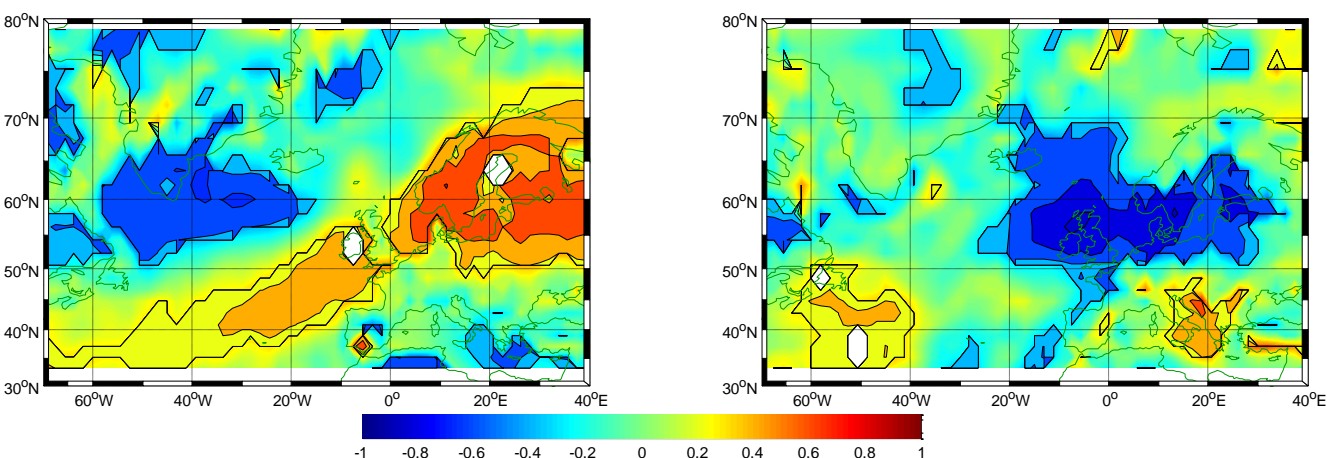

**Figure 14: The correlation patterns based on de-trended time series between the NEF(+↓) and the BANOS-index in wintertime (left) and summertime (right) over the period 1949-2013. For this record length, the 95% significance level is ±0.24. The areas indicating significant correlations are delineated with contour lines.**

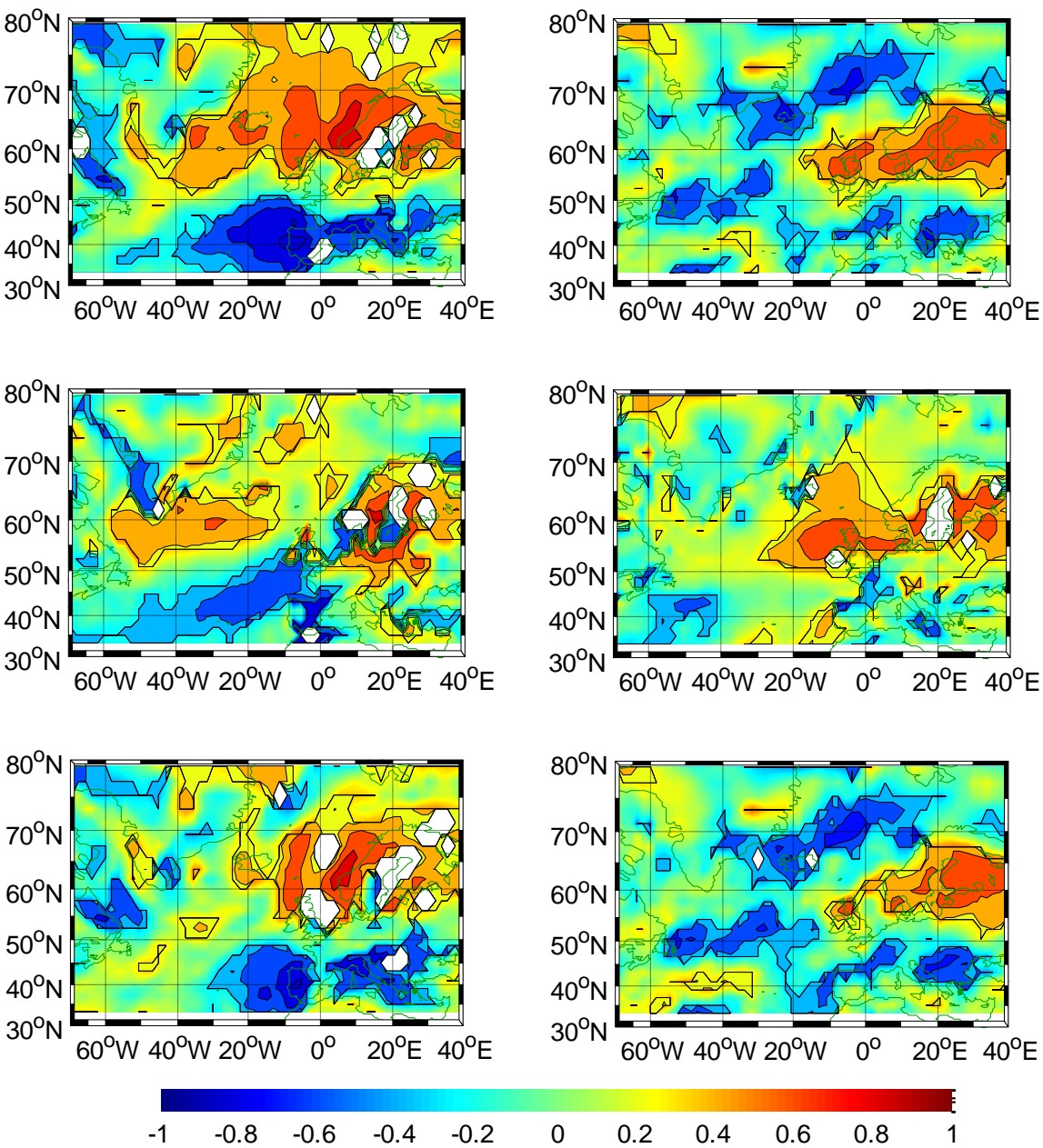

**Figure 15: The correlation patterns of precipitation (top), evaporation (middle) and freshwater flux (bottom) to the BANOS-index in wintertime (left) and summertime (right) over the period 1949-2013. Time series are de-trended prior to the correlation computation. The 95% significance level is ±0.24. The areas indicating significant correlations are delineated with contour lines.**

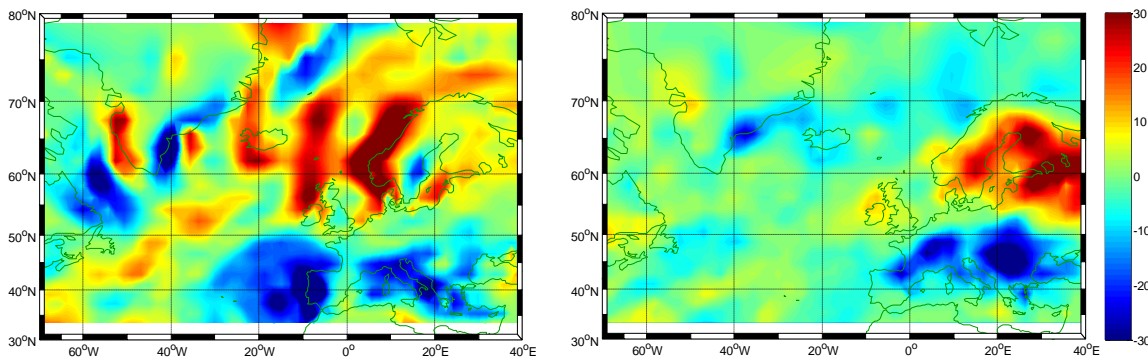

**Figure 16: The sensitivity patterns of the freshwater flux grids per unit change in the BANOS-index in wintertime (left) and summertime (right).**

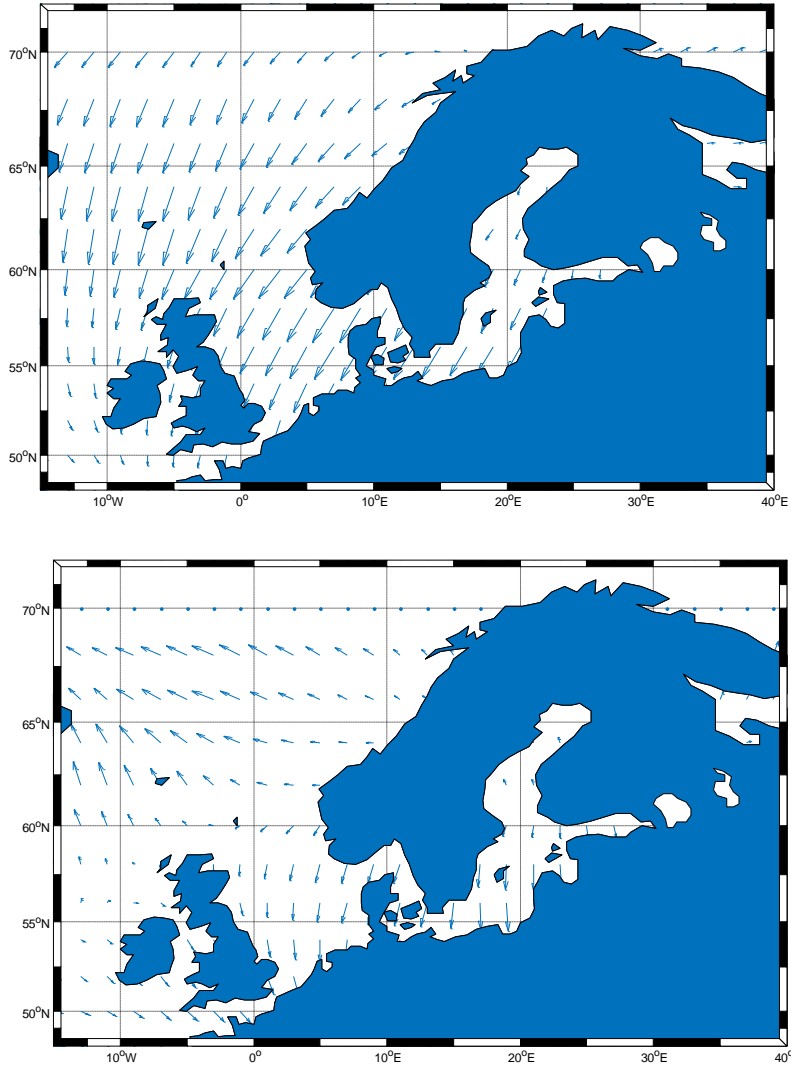

**Figure 17: The vectors represent the direction of the Ekman transport caused by the SLP BANOS patterns, assuming a geostrophic wind aproximation and a complete ocean Ekman layer. The lengths of the vectors are proportional to the magnitude of the surface wind, and directed 90 degrees to the right of the surface wind. Upper (lower) panel shows the winter (summer) season Ekman transport vectors.**