# Peer review of "Contribution of atmospheric circulation to recent off-shore sea-level variations in the Baltic Sea and the North Sea"

_Earth System Dynamics, 2017_

## Referee Comment (RC1) · Anonymous Referee #1 · 13 Apr 2017

**General**

The authors propose a new proxy index for large-scale pressure variability over Western Europe that explains a large fraction of sea level variability. They argue that this index, which is named the 'Baltic Sea and North Sea Oscillation'  and  corresponds to the air pressure difference between mid-Scandinavia and the Bay of Biscay, explains a larger fraction of sea level variability in the Baltic Sea and the German Bight, compared to the North Atlantic Oscillation index. Using a regression analysis, they argue that the inverted barometer effect explains the majority of the observed correlation pattern, and that the net energy and freshwater flux though the sea surface explains an other part. Finally they argue that Ekman transport generated from the atmospheric pressure pattern and the induced winds do not result in water exchange between the North Sea and the Baltic Sea in wintertime.

This subject is interesting, as this variability affects long-term trends and have an impact of sea level extremes. However, in my feeling, there are still substantial shortcomings, which makes the conclusions not yet convincing. I doubt whether the proposed index is novel, as the pattern has been found before, and I think the analysis of the results should go beyond showing correlation patterns, which do not prove the direction of causation. Hence, in my opinion, the paper is not publishable in its current form.

**General comments**

1. Dangendorf et al. 2013 found a very similar correlation pattern between sea level at the Cuxhaven tide gauge and atmospheric pressure. Furthermore, Dangendorf et al. 2014a derived an atmospheric proxy for sea level variability in the North Sea, based on the pressure difference from virtually the same areas to what has been presented here. Does the BANOS proxy represent a different pattern and does it perform better than this proxy? Otherwise, this study seems of limited use.
2. The results almost only show correlation patterns. These patterns can give insight, but it does not show the amplitude of the signals involved. Which fraction of the observed sea level variability can be explained by the atmospheric proxy? What about the fraction of explained variance (R-squared) as a measure of the BANOS model skill? Xu et al. 2015 show that the typical amplitude of variability differs widely within the Baltic (See their figures 3 and 4). Does the coherent NAO/BANOS-induced variability share a coherent basin-mean signal, only with regionally-varying amplitude?  What is the standard deviation/RMS of the residual sea level after removing the NAO/BANOS signal?
3. What are the typical time scales of the variability explained by the BANOS index? Are we explaining monthly variability, seasonal, annual or even longer variability? The abstract suggests 'interannual', which is sometimes repeated, but is not worked out. Since many processes act on different time scales, this classification is very necessary. For example, North Sea  variability on decadal time scales is generally assumed to be driven by integrated longshore winds that cause coastally-trapped waves (See Calafat et al. 2012/2013, Dangendorf et al. 2014b and Frederikse et al. 2016), which has not much to do with NAO/BANOS-related effects. Does this signal affect the Baltic Sea? A tool that can be suitable to find the relevant timescales at which the correlations are largest is the wavelet toolbox from Aslak Grinsted (http://www.glaciology.net/wavelet-coherence). Furthermore, a plot that shows observed sea level and the fraction explained by the BANOS index could give more insight.
4. It's not clear to me how the time series are formed: do the authors use a mean value for each summer/winter (thus one value per year), or do they use the monthly data from the winter/summer months (thus multiple values per year)? How is the seasonal cycle treated?
5. The region is unique due to the presence of many long tide gauge records. Why not use all of these records to show the capability of the BANOS index? Figure 6 and 7 suggest a non-uniform NAO response at different tide gauge locations. This analysis may also provide the much-needed insight into my points 2 and 3 above. Furthermore, the analysis of long-term records in the North Sea only seems to cover the German Bight, while many more tide gauges are available for most of its coastlines.

6. The inverted barometer effect (IBE) is and only is the static sea level response to air pressure anomalies, and therefore dictates a fixed regression coefficient of -1 cm/hPa. Therefore, the observed pressure correlations, for which a different regression coefficient is found are not only resulting from IBE.

7. The conditions for Ekman transport to develop are to my knowledge not satisfied, I'd guess, since the Baltic Sea is small and very shallow. For Ekman transport to develop, the dominating balance in the equations of motion is between wind stress and the Coriolis force. Since the Baltic Sea is both shallow and small, bottom friction will probably play a large role, and the basin size is probably smaller than the Rossby radius of deformation. Hence, I'm not convinced by the conclusions that follow from this analysis. Many other studies point at the large influence of local winds on sea level variability here. It may be worthwhile to investigate the wind effects using a barotropic ocean model. These models can explain a large fraction of the observed sea level variability of monthly-mean data, as shown by Xu et al. (2015).

8. In a paper by Chen et al [2014], the role of barotropic and baroclinic responses to the NAO in the North Sea are extensively discussed. One of their main arguments is that local density effects on a shallow shelf are small, but a horizontal pressure gradient that develops when a deep ocean column expands results in mass transport towards the shelf. How could we combine these results with your attribution process, which relies quite heavily on density effects (freshwater flux/heating)? Are in-situ temperature and salinity profiles available in the region to verify whether local density effects play a substantial role? Otherwise, regional ocean reanalysis products (some are available at http://marine.copernicus.eu) may provide estimates. From the observation of the anti-correlation between BANOS/NAO and open-ocean sea level, couldn't it be a wind-driven mass redistribution process? Over the last 15 years, you may have a look at what GRACE observations say about mass changes.

**In-line comments**

Note that the page numbering re-starts every page. I use the PxLy notation, referring to page x line y.

Title and P1L6: This study mainly deals with the Baltic Sea, and only partially with the North Sea. I suggest: 'German Bight' instead of North Sea

P4L15: References to Dangendorf et al. 2013/2014 should be discussed here, and further on, what do we learn from this paper that we do not know yet after reading these papers?

P5L3: For completeness, it's a good idea to add links to the web sites from which you've obtained the data.

P5L11: Do you derive season-means from monthly data? Or monthly data only over this period? What about spring and autumn?

P5L13: Altimetry data does not have a ¼ by ¼ degree resolution: along-track observations are interpolated onto a grid which can have a higher resolution than the data from which it is composed. Note that observations are integrated over distances of about 100 km (See Le Traon et al 2001 or Pujol et al. 2014). Furthermore, observations deteriorate quickly close to land, and shallow-water tides may alias into lower frequencies. Hence, it it very tricky to separate small-scale features in shallow shelf seas. Tide gauges are generally more reliable in such areas. An alternative may be to use along-track altimetry observations, which do not suffer from problems related to interpolation. These are widely available from AVISO.

P6L2: The word 'slope' here seems a bit misleading: you compute the regression coefficient between the atmospheric pattern and sea level. 'Slope' suggests a linear trend to me.

P6L11: The NCEP/NCAR reanalysis 1 is not really state-of-the-art anymore. Furthermore, since you use this data set to derive heat fluxes and precipitation over sea, how good does this model perform for these quantities? I guess that this model does not directly assimilate heat flux and precipitation data, and that they are derived from wind and pressure data. It may be worthwhile to use something like MERRA or ERA-interim, in which flux observations derived from satellites are assimilated. An other alternative may be OAflux (http://oaflux.whoi.edu/)

P6L34: This statement seems easy to verify: what is the correlation over the common altimetry/TG period?

P7L10: Why only check for these three? If you use all available tide gauges in the region with a substantial amount of observations, you can generate a map with the correlations at each TG location. This will make much clearer whether altimetry observations do a good job, especially at the narrow straits (Kattegat/Skagerrak etc) and around islands.

P7L11: Could you make the followed procedure more clear? I don't fully understand how the data has been treated. I also wonder how you treated the seasonal cycle.

P7L16: A correlation only does not show that signals are coherent: what about the fraction of explained variance, or a simple plot, in which both time series are compared?

P7L17: Are you sure that you compare the same signals with Yan et al [2004] ? Same treatment of seasonal cycle/lowpass filters/detrending etc? Maybe this statement falls outside the scope of this manuscript.

P7L24: From this correlation pattern I'd assume that the tide gauge records have a high mutual correlation. Could you show the time series of all tide gauges in one plot to verify this?

P7L28: This is an interesting finding and may give some hints about the underlying processes!

P8L28: How do you define 'significant'?

P9L3: Add here very explicitly about which time scales you are discussing.

P9L7: 'tend to be spatially coherent': again, the large amplitude variations over the region do not support this statement.

P9L8: 'the influence of NAO is spatially quite heterogeneous': where do you show this?

P9L26: These findings contradict with the conclusion that Baltic sea level is coherent, as the difference in the found correlations is rather large, which would not be the case if the signals at both tide gauges were coherent.

P10L19: The correlation patterns seem to be almost the same as already found by Dangendorf et al, 2014a. Following the argumentation, this makes sense, as sea level variability in the Southeastern North Sea, used by Dangendorf et al, and the Baltic sea is coherent and thus has a common driver. Can't we just suffice by saying: "The atmospheric proxy, developed by Dangendorf et al. [2014] does not only work as a proxy for the S.E. North Sea, but also for the Baltic Sea."? Or is the new BANOS index doing significantly better? The only difference between Dangendorf et al and BANOS seems to me the Eastward shift in summer for the BANOS index. However the summer correlations are substantially less convincing than in winter, as I observe in figure 10. Especially the right panel in fig. 10. How does the model perform when you stick to the winter definition, even for summer?

P11L11: It seems that there are some decadal features that are shared between NAO and BANOS. Here, a wavelet analysis, as described above, may be more insightful.

P12L8: Isn't this negative trend just a symptom of the non-stationary correlation? Something similar happened between 1905-1935.

P12L9: I'd say: "No significant link between NAO and Baltic sea level in summertime"

P12L11 and the following section: Like the NAO, you find a strong anti-correlation between North Atlantic sea level and the BANOS index. That's an interesting finding in my opinion.

P13L5 and the following sections: Here you show the spatially heterogeneous senitivity, again pointing at a spatially varying sea level signal. Which fraction of the variability is explained?

P13L9: Avoid the word 'slope' here. Maybe insert a short equation:
dSLA = a*BANOS with a in [mm/BANOS]

P13L12: It may well be the case that next to BANOS, more effects are at play here, that are not necessarily linear.

P14L4: Why is that suggested? Horizontal pressure gradients will result in a sea level gradient due to the IB effect, and generate geostrophic winds. Do you mean that wind effects play a role?

P14L18: As stated in the introduction: The IBE effect is the static response to pressure effects with a sensitivity of 10 mm/hPa. Since, static equilibrium is generally reached on timescales in the order of days, deviations from this static effect imply that some other effects are at play here. That's not so strange, as close to coastal areas, winds play a large role. To separate these effects, a barotropic ocean model can bring more clarity.

P15 equation 1: I'd suppose that the rate of change in steric sea level correlates with the heat flux and not the sea level itself. i.e. dSL/dt ~ Q_net instead of SL ~ Q_net. What if you integrate Q_net before computing any correlation?

P15L24: This number is rather large, I suspect it's incorrect.

P15L25 and below: How did you compute this? If I'd compute the thermal heating that result of the  aforementioned number, the whole Baltic sea would evaporate rather quickly. Are there in-situ T/S observations, or SST observations that can confirm the large impact of density changes induced by local heat fluxes? I'd guess that on a shallow shelf, the effect of density changes is rather limited. Furthermore, if so much water evaporates or rains into the basin, doesn't the resulting sea level change lead to transport with the open ocean?

P16L11 and beyond: Like with energy fluxes, isn't it expected that sea level varies according to the integral of the total freshwater flux?:
dSL/dt ~ E
Furthermore, do you suppose that the total mass in the Baltic Sea is affected, or that the effects are caused by changes in the salinity?

P16L25: The correlation pattern does not tell much about what causes what. I'd say that the precipitation/evaporation pattern changes and sea level changes are both caused by the BANOS-related pressure changes. Therefore, they show mutual correlation. But that does not show that the P-E flux causes sea level changes! Hence, the conclusions reached from P16L32 onward are not really justified in my opinion without further proofs.

P17L7: Do you mean 'geostrophic wind flow' here?

P17L8: I don't understand what you mean here: why can't the BANOS-induced wind forcing transport surface water between both basins? Although for different regions,  many studies point at the large impact of local wind variations on monthly and interannual sea level, including Sterlini et al. 2016, Dangendorf et al. 2013, 2014a, 2014b, and many more. Does the same happen in the Baltic Sea? Figure 3a in Dangendorf et al, 2014a clearly suggest a wind set-up effect.

P17L15: To my knowledge, as described above, both the local bathymetry and shallow water, as well as the presence of coasts render the Ekman transport assumption invalid. The width of the basin is about 100 – 200 km, which is probably smaller than the Rossby radius of deformation. Especially around Skagerrak and Kattagat, the basin dimensions become very small. Hence, I would not trust results based on the Ekman transport assumption. Again, a wind-forced barotropic ocean model or a regionalocean reanalysis could bring more trustworthy results regarding changes in wind-induced transport and sea level variability.

P18L3: The impact of NAO/BANOS-related variability is the only atmospheric effect on sea level analysed in this study. There may be more atmospheric processes affecting sea level on interannual time scales.

I'm afraid that most conclusions are not justified by the presented results:

- Conclusion #1: According to other studies, the amplitude of the interannual variability differs widely around the region and therefore, the variability is not coherent. Furthermore, in figure 14 you show that the spatial signal is far from coherent, as the sensitivity values differ by a factor 10 over the basin.
- Conclusion #2: In figure 6, top I see a rather strong correlation between the NAO and the altimetry-derived sea level in wintertime, rather than weak!
- Conclusion #3: There are essentially two indices: summer-BANOS and winter-BANOS.
- Conclusion #4: The BANOS index only correlates with sea level variability in the German Bight, and not in the whole North Sea in summer.
- Conclusion #5: Since the regression coefficient deviates from the static IB response, it's probably not only the IB effect that causes the pressure-sea level link. Furthermore, if the IB effect explains 88 % of sea level variability, and surface fluxes 35%, we explain more than 100 %. Again, you've only showed a correlation pattern and not what causes what. They also may have a common cause. How are these percentages derived? The conclusions regarding wind-driven variability depend on the Ekman transport approximation, which is probably not valid in this region.

Figures,

1. In general, it may be a good idea to avoid the 'rainbow' color scale for correlations. A good summary of which color maps are suitable can be found here: https://betterfigures.org/2015/06/23/picking-a-colour-scale-for-scientific-graphics/ I'd suggest to use a 'diverging' color scale. It looks like you use GMT for the plots, for which many good diverging color palettes can be found here: http://soliton.vm.bytemark.co.uk/pub/cpt-city/jjg/cbcont/div/index.html

2. It may also be a good idea to contour areas with significant correlations

3. Some figure captions can be expanded to describe the followed procedure. For example figure 4: "The correlation pattern between de-trended sea level during the winter months(DJFM) and the de-trended NAO index over the same months. The correlation has been computed between January 1993 and December 2014" or something similar. This will allow easier reproduction of your results.

Figure 1: Maybe add the locations of the tide gauges

Figure 10: On the left, some data seems to be missing

**References**

Calafat, F. M., Chambers, D. P., & Tsimplis, M. N. (2012). Mechanisms of decadal sea level variability in the eastern North Atlantic and the Mediterranean Sea. Journal of Geophysical Research: Oceans, 117(C9).

Calafat, F. M., Chambers, D. P., & Tsimplis, M. N. (2012). Mechanisms of decadal sea level variability in the eastern North Atlantic and the Mediterranean Sea. Journal of Geophysical Research: Oceans, 117(C9).

Chen, X., Dangendorf, S., Narayan, N., O'Driscoll, K., Tsimplis, M. N., Su, J., ... & Pohlmann, T. (2014). On sea level change in the North Sea influenced by the North Atlantic Oscillation: local and remote steric effects. Estuarine, Coastal and Shelf Science, 151, 186-195.

Dangendorf, Sönke, et al. "Characteristics of intra-, inter-annual and decadal sea-level variability and the role of meteorological forcing: the long record of Cuxhaven." Ocean Dynamics 63.2-3 (2013): 209-224.

Dangendorf, S., Wahl, T., Nilson, E., Klein, B., & Jensen, J. (2014a). A new atmospheric proxy for sea level variability in the southeastern North Sea: observations and future ensemble projections. Climate dynamics, 43(1-2), 447-467.

Dangendorf, S., Calafat, F. M., Arns, A., Wahl, T., Haigh, I. D., & Jensen, J. (2014b). Mean sea level variability in the North Sea: Processes and implications. Journal of Geophysical Research: Oceans, 119(10).

Frederikse, T., Riva, R., Kleinherenbrink, M., Wada, Y., Broeke, M., & Marzeion, B. (2016). Closing the sea level budget on a regional scale: Trends and variability on the Northwestern European continental shelf. Geophysical Research Letters, 43(20).

Le Traon, P. Y., Dibarboure, G., & Ducet, N. (2001). Use of a high-resolution model to analyze the mapping capabilities of multiple-altimeter missions. Journal of Atmospheric and Oceanic Technology, 18(7), 1277-1288.

Pujol, M. I., Faugère, Y., Taburet, G., Dupuy, S., Pelloquin, C., Ablain, M., & Picot, N. (2016). DUACS DT2014: the new multi-mission altimeter data set reprocessed over 20 years. Ocean Science, 12(5), 1067-1090.

Sterlini, P., de Vries, H., & Katsman, C. (2016). Sea surface height variability in the North East Atlantic from satellite altimetry. Climate Dynamics, 47(3-4), 1285-1302.

Xu, Q., Cheng, Y., Plag, H. P., & Zhang, B. (2015). Investigation of sea level variability in the Baltic Sea from tide gauge, satellite altimeter data, and model reanalysis. International Journal of Remote Sensing, 36(10), 2548-2568.

Yan, Z., Tsimplis, M. N., & Woolf, D. (2004). Analysis of the relationship between the North Atlantic oscillation and sea-level changes in northwest Europe. International Journal of Climatology, 24(6), 743-758.

---

## Referee Comment (RC2) · Anonymous Referee #2 · 28 Apr 2017

General Comments

The paper at hand offers an original approach on how seasonal mean sea level in the Baltic Sea are related to large scale atmospheric patterns. The analysis is based on correlations between sea level pressure differences and tide gauge and altimeter derived sea level anomalies. An effort is made to come up with explanations on how different atmospheric processes might contribute to the sea level variability.

A new index is introduced that overcomes the poor correlation between the NAO-index and sea levels in Baltic Sea during summer. Also, the varying degree of long term correlations between NAO-index and winter mean sea levels in the Baltic Sea has been addressed. The most important contribution of the paper is probably to renew

the discussion on how seasonal mean sea levels in the Baltic Sea are related to the regional climate.

I definitely recommend the paper for publication. Some parts are weaker than others and those should be reformulated before final publication. The paper also might be improved by addressing the specific questions and comments below.

Specific Comments

A major part of the paper is the presentation and discussion of how much better the BANOS-index correlates to sea level changes in the Baltic Sea than the NAO-index. It would be helpful to the reader to discuss the BANOS-index with respect to other slp indices that have been used for the Baltic Sea. For example the BAC index in Andersson, 2002 or the BSI in Lehmann et al, 2002. How much different are those indices from the BANOS-index? Would the slp gradient over the transition zone between North Sea and Baltic Sea be different between BANOS and BAC or BANOS and BSI?

Is the BANOS-index usable without a gridded slp field? Can it be inferred from station data (e.g. Stockholm - Odessa), like the NAO-index and would it show good correlation the the one derived from gridded data?

How sensitive are the results and the conclusions for summer when the summer was defined as the months JAS? Baltic Sea sea level in summer show little variation during these three months (e.g. Hünicke and Zorita, 2008, Meier et al., 2004). The month of June usually lies between the spring minimum and the summer values. What is the reason to choose June, July and August as the summer season?

The title of the paper suggests the investigation of North Sea and Baltic Sea sea levels. In the derivation and evaluation of the BANOS-index only Baltic Sea sea levels are taken into account. From Figures 4 and 13 one would expect that the correlation of winter mean sea levels in the North Sea improve over the NAO correlations. For summer at least in the eastern part of the North Sea. Please consider to show one or

two tide gauges from the North Sea in Figure 12, or add a figure like Figure 12 for two North Sea tide gauges.

In the abstract it is said that the wind associated with the slp pattern of the BANOS-index does not lead to transport of water into the Baltic Sea. The analysis in Section 4.8.4 is a good idea, but since the return flows along the coasts and at depth are not taken into account, the statement might not hold up to a more thorough investigation.

I would like to see the correlation coefficients for all of the nine stations in Table 1. Future studies might benefit from that information and for the present study the information might help with the interpretation of Figures 2 and 3.

Good correlations between altimetry and tide gauge data is said to indicate progress in satellite altimetry. Could the source of the improvement over earlier comparisons be specified? It is probably not the altimeter instruments themselves that have progressed so much. Information on the geoid? New algorithms? Amount of data?

In Section 4.8.1 yet another index is introduced as the slp differences between two geographical locations that differ from the ones introduced in Section 4.4. Would you please explain why there is a need to introduce different locations where to measure slp to get at the IBE contribution? Why Denmark Strait and not Labrador Sea for the summer for example? But more importantly why not stick to the definition (5W, 45N) - (20E, 70N) for winter and (30E, 45N) - (20E, 60N) for summer?

In Section 4.8.2 please indicate in the text whether the whole water column was heated or just the mixed layer (down to what depth?) to absorb the energy.

Section 4.8.4 discusses an interesting point but the argumentation stops halfway through. How is the Ekman transport different in the transition zone between North Sea and Baltic Sea for NAO- or BANOS-index related patterns? And what are the consequences for the sea-level in the Baltic Sea?

In Section 4.8.4 it is argued that during summer BANOS and NAO related wind forcing

(slp gradients) could be similar. I thought the main reason to introduce the BANOS index was the liberty to define an index separately for winter and summer, so that missing correlation for summer sea-level variability in the Baltic Sea could be explained. This section would benefit from a rewrite, I suppose.

The Conclusions (page 18, line 28-29) mention that there is no contribution of NEF in summer. Section 4.8.2 states that summer has not been included in the analysis, because of negative correlations. Please reformulate the sentence in the conclusions, page 18, line 28-29.

For the last part of the Conclusions (page 19, line 2-4) the evidence is missing. See also the comments above on Section 4.8.4 and the Abstract.

The conclusion might benefit from a restructuring. The itemized list could be shortened to contain the keywords only as a summary of a continuous text around it.

Technical Corrections

page 1, line 7: interannual time scales.

page 2, line 24: NAO-index describes weaker

page 3, line 19: Andersson (2002) who focused on

page 3, line 20: Dangendorf et al. (2014) who investigated the

page 3, line 20-21: investigated the North Sea, reported that atmospheric variability that differs from the NAO may still explain part of the sea-level variability.

page 3, line 32: different sorts of data sets

page 4, line 32: This study focuses on the winter

page 5, line 1: this study and the following section

page 5, line 5-7: Although the threshold of any computation involving tide gauge records was set at 75% availability of data for the considered period, seasonal means

are calculated in case of availability of tide gauge records for two months. [I am not sure whether I understand it the way the authors mean it. Could this information be written a little more clearly?]

page 5, line 23: the northeastern boundary of [If Smögen is meant by the northeastern boundary of the North Sea I recommend to call it Skagerrak.]

page 6, line 14: and produces a complete gridded data set

page 6, line 16: resolution of a T62 Gaussian grid [Or mention the resolution in deg or km]

page 7, line 9: examine the correlation of the satellite [see below]

page 7, line 11-13: The seasonal means ... linear regression. [This has been said in the methods section. It could be scratched.]

page 8, line 30, 32 in the southern Baltic Sea

page 9, line 5: Figure 4-lower [see also comment below]

page 9, line 28: does seem to be strongly connected to the NAO-index from 1998 in wintertime. OR does not seem to be strongly connected to the NAO-index until 1998 in wintertime.

page 9, line 30: could indicate the existence of

page 10, line 30: from what the NAO implies.

page 11, line 33: most of the time the gliding

page 13, line 31: region on the interannual time scale. OR region on interannual time scales.

page 14, line 27: (3.44*18.09) OR (3.44 \cdot 18.09) AND (1.39*7.02)

page 15, line 3: radiation reaching the earth surface and longwave (LW) emitted

page 15, line 9: [SH and LH do not correspond to the naming in the equation.]

page 15, line 18: estimated as 3.28 (W m-2 u-1)

page 15, line 24: of 25,505,280 (W m-2) per one unit [How much change is that? It might be easier to grasp by using something like 2.5 10ˆ7 (W m-2)]

page 15, line 24: of 25,505,280 (W m-2) per one unit [Should the units be (J/winter m-2) instead of (J/s m-2)?]

page 16, line 9: (2257 kJ kg-1)

page 16, line 16-17: [During summer the western North Sea is not similar in precipitation and freshwater flux.]

page 16, line 29: [It would be more concise to use the regions defined in Figure 1 when describing the correlation pattern.]

page 17, line 2: of sea-level would reach $\sim$ 10 mm per one unit

page 17, line 3: [Either remove the last sentence or explain why the high correlation in the north eastern part of the North Sea drainage basin (incl. Norway) does not contribute to North Sea sea level variability.]

page 17, line 14: be similar to the case of the NAO.

page 17, line 15: assuming that the Ekman layer

page 17, line 16: interrupted by bathymetry.

page 17, line 21: if bathymetry would not interrupt

page 17, line 22: is generated by the BANOS-related north-easterly winds

page 18. line 1: [The Norwegian coastline is in the north-eastward direction. That contradicts the previous argumentation.]

page 18, line 4: on interannual time scales. OR on the interannual time scale.

page 18, line 15: is more important for the

page 18, line 32: [The last sentence seems to contradict what has been said in the first sentence (line 30-32). Would you please specify which contributions of the freshwater flux are negligible.

page 19, line 4: to that related to the NAO in

caption Figure 14: Note the different intervals on the color scales.

Readability: The readability of Section 2 could be improved by omitting the sub-sub-sections.

Figure captions: I'd recommend to change the naming of the positions in the plots from down to lower/bottom and from up to upper/top.

Coherency or correlation (page 7, line 9ff): Did you really look at correlations on different time scales or is "coherence" used here as a synonym for correlation? The expression coherence or coherency appears more often later on. It would be helpful to specify what is meant by coherency or perhaps just use correlation.

Station Smogen (page 5, line 24ff): It appears in some places in the text, it should be Smögen.

Caption Figure 5: for the winter (solid line) and summer (dotted line) seasons.

Figure 8-9: These two figures could be combined in the same way Figure 10 was done.

Figure 10: Why is the northern half of the figure blank? If possible, the figure should be redone. It avoids unnecessary doubts.

Figures 8-10: The reader might appreciate the eye be guided with a mark on the plots, where the BANOS-index is defined.

Figure 11: It is hard to see similarities and even more so difference between the two indices. An additional running mean could improve the figures. Red-blue bar plots like

they are used for the display of the NAO-index might be an option.

Caption Figure 18: It is not clear which index or which season is displayed in the upper and lower panel of Figure 18. Also, for the sake of completeness it would be good to indicate the units and a reference vector.

Sub-sub-section 4.8.2: It would be clearer to use one name only for the net energy flux. Either Qnef or NEF. Also, in the text the "net"- and "nef"-part of names like SWnet should be a subscript. Or maybe drop the "net" altogether in the whole section.

Sub-section 4.8: In my opinion it would suffice to indicate the sensitivities with one digit after the floating point.
* * *

---

## Author Comment (AC1) · 13 Jun 2017

We thank the reviewer very much for reviewing our manuscript, for providing constructive criticism and useful suggestions. We respond to all comments below.

**General comments**
*1.Dangendorf et al. 2013 found a very similar correlation pattern between sea level at the Cuxhaven tide gauge and atmospheric pressure. Furthermore, Dangendorf et al. 2014a derived an atmospheric proxy for sea level variability in the North Sea, based on the pressure difference from virtually the same areas to what has been presented here. Does the BANOS proxy represent a different pattern and does it perform better than this proxy? Otherwise, this study seems of limited use.*

The BANOS mode of atmospheric circulation indicates different atmospheric pattern than SLP pattern that Dangendorf et al. 2014a suggest, especially in summertime. This difference also affects the role of the physical factors that explain the linkage between BANOS-index and sea-level variability. For example, in our interpretation, the BANOS mode does not indicate a wind-driven surface water transport from the North Sea to the Baltic Sea over the transition zone in wintertime. Furthermore, the correlation pattern (Figure 13-top) between the BANOS-index and SLA grids indicates a large scale seesaw effect of BANOS mode on sea-level variability between the North Atlantic and the Baltic Sea region. Our study explains that the Inverse Barometer Effect (IBE) plays a key role explaining the linkage between SLP BANOS pattern and sea-level variability. This effect was not discussed in the previous studies (i.e. Andersson 2002 and Dangendorf et al. 2014a).

To illustrate the covariability between those two indices (BANOS and Dangendorf et al.2014a proxy), we computed the correlation coefficients of de-trended time series between those two indices. For wintertime (summertime) the correlation is 0.89 (0.74) for the period 1900-2008. The standardized index time series (not de-trended) in the winter(upper panel) and summer seasons for the period 1900-2008 are shown below.

[Figure]

*2.The results almost only show correlation patterns. These patterns can give insight, but it does not show the amplitude of the signals involved. Which fraction of the observed sea level variability can be explained by the atmospheric proxy? What about the fraction of explained variance (R-squared) as a measure of the BANOS model skill? Xu et al. 2015 show that the typical amplitude of variability differs widely within the Baltic (See their figures 3 and 4). Does the coherent NAO/BANOS-induced variability share a coherent basin-mean signal, only with regionally-varying amplitude? What is the standard deviation/RMS of the residual sea level after removing the NAO/BANOS signal?*

In Figure 13, we show the correlation pattern between BANOS-index and SLAs for wintertime and summertime. Actually, those figures provide information about the fraction of sea-level variability that can be explained by the BANOS-index. For this computation, correlation coefficients should be squared. Additionally, we mention about the BANOS explained variance of sea-level in the different parts of the manuscript (i.e. "Abstract", Page 1 Line 18-20).

To show the amplitude of the sea-level variability involved in the BANOS mode of atmospheric circulation, we computed the sea-level standard deviations from observations, from the sea-level explained by BANOS-index predictions and from the residuals. For the BANOS prediction, we applied a linear regression between BANOS (predictor) and satellite SLAs (predictand) for each SLA grid over the period 1993-2013. The residuals are deduced from that linear regression. The results are provided in the following figure (units: mm).

[Figure]

Figure shows the amplitude of sea-level variability from observations (top), BANOS predictions (middle) and the associated residuals (bottom) (Measured SLAi – BANOS predicted SLAi) for the winter (left panels) and summer (right panels) seasons for the period 1993-2013. In the figure, the scale is up to 180 (60) mm for the winter (summer) season.

We note that observations and BANOS predictions depict consistent spatial distribution of the standard deviations, especially in wintertime. For wintertime, high sea-level standard deviations occur in Bothnian Bay, in Baltic proper, in the Gulf of Finland and in the Gulf of Riga. The

residuals show relatively small and spatially homogeneous standard deviation (~50 mm) distribution in the Baltic Sea and the North Sea in wintertime. In summertime, the spatial distribution characteristics are also consistent. For summertime, the Gulf of Riga shows the highest standard deviation, which is also showed by the standard deviations explained by the BANOS index. Again in summertime, it could be said that residuals show small standard deviation values (>25 mm) and a homogeneous pattern in the Baltic Sea basin except for the Bothnian Bay and the southern part of the North Sea where the standard deviations differ.

Overall, the results indicate that BANOS-induced atmospheric signal can explain a considerable amount of sea-level variability in the Baltic Sea and North Sea region. Especially in wintertime, the BANOS-index explains almost of all sea-level variations linked to the atmospheric circulation.

*3.What are the typical time scales of the variability explained by the BANOS index? Are we explaining monthly variability, seasonal, annual or even longer variability? The abstract suggests 'interannual', which is sometimes repeated, but is not worked out. Since many processes act on different time scales, this classification is very necessary. For example, North Sea variability on decadal time scales is generally assumed to be driven by integrated longshore winds that cause coastally-trapped waves (See Calafat et al. 2012/2013, Dangendorf et al. 2014b and Frederikse et al. 2016), which has not much to do with NAO/BANOS-related effects. Does this signal affect the Baltic Sea? A tool that can be suitable to find the relevant timescales at which the correlations are largest is the wavelet toolbox from Aslak Grinsted (http://www.glaciology.net/wavelet-coherence). Furthermore, a plot that shows observed sea level and the fraction explained by the BANOS index could give more insight.*

In this study, we analyse sea-level variability on interannual time scale. All time series involved in the analysis are winter means(December-January-February) and summer means (June-July-August), which are computed from monthly means.

Considering the analysis technique that Grinsted used, we applied a frequency domain analysis (Fourier Analysis) on the Stockholm sea-level and BANOS-index time series for winter seasonal means. The power spectrum of BANOS-index (upper panel) and of the Stockholm record in the following figures. Time series are detrended prior to the analysis and the analysis period was 1900-2013.

[Figure]

The time series show a white noise character, with no clear peaks in the spectrum.

In addition, several researchers using different techniques have examined the power spectra of the NAO indices. A spectral analysis on the NAO-index (Hurrell et al. 2003) indicates that the spectrum of winter mean NAO index is red, but there is no significant peak.

*4.It's not clear to me how the time series are formed: do the authors use a mean value for each summer/winter (thus one value per year), or do they use the monthly data from the winter/summer months (thus multiple values per year)? How is the seasonal cycle treated?*

For the whole analysis, we used seasonal mean value for each winter and summer. Those winter and summer mean values are computed from monthly means prior to the analysis. We analysed winter and summer separately. Thus, there is no need to remove the seasonal cycle. We will make some clarification in the manuscript about it.

*5.The region is unique due to the presence of many long tide gauge records. Why not use all of these records to show the capability of the BANOS index? Figure 6 and 7 suggest a non-uniform NAO response at different tide gauge locations. This analysis may also provide the much-needed insight into my points 2 and 3 above. Furthermore, the analysis of long-term records in the North Sea only seems to cover the German Bight, while many more tide gauges are available for most of its coastlines.*

Keeping in mind that interannual sea-level variability in the Baltic Sea and the North Sea is spatially quite coherent (i.e. Stramska 2013), we used nine representative tide gauges assuming that they would be representative for sea-level variability on interannual time scale in this region.

The correlation pattern between BANOS-index and satellite SLAs indicates that only the eastern part of the North Sea(only German Bight) is connected to the BANOS mode of the atmospheric variability in wintertime(summertime). Since Dangendorf et al. (2013) considered the Cuxhaven record to analyse sea-level variability in the German Bight, we also carried out a statistical analysis considering the connection between the BANOS-index and the Cuxhaven station. That statistical analysis indicates that 64% of sea-level variance can be explained by the BANOS-index in wintertime for the period 1900-2008. The following table show the correlation coefficients among the BANOS-index, the Cuxhaven and Stockholm stations for the period 1900-2008.

| Corr. Coeff. Winter (Summer) | Cuxhaven | Stockholm |
|---|---|---|
| BANOS | 0.80(0.50) | 0.84(0.72) |
| Cuxhaven | - | 0.88 (0.67) |

*6.The inverted barometer effect (IBE) is and only is the static sea level response to air pressure anomalies, and therefore dictates a fixed regression coefficient of -1 cm/hPa. Therefore, the observed pressure correlations, for which a different regression coefficient is found are not only resulting from IBE.*

The comment by the reviewer assumes that air pressure changes only at the location of the tide gauge. In the manuscript, we will clarify that the inverse barometer effect (IBE) should include the pressure gradients, since water should be transported from one point where pressure increases to another where pressure decreases. We assume the presence of an infinite ocean without topographic limitation and complete equilibrium in the Baltic Sea and North Sea region. For example, concerning the winter season, the increase in high SLP system over the North Atlantic (especially around Gulf of Biscay- Figure 9) pushes the water into region where the low-pressure decreases in the Baltic Sea and North Sea region.

We will rewrite that part of the text in order to make the explanation more clear than in the current version.

*7.The conditions for Ekman transport to develop are to my knowledge not satisfied, I'd guess, since the Baltic Sea is small and very shallow. For Ekman transport to develop, the dominating balance in the equations of motion is between wind stress and the Coriolis force. Since the Baltic Sea is both shallow and small, bottom friction will probably play a large role, and the basin size is probably smaller than the Rossby radius of deformation. Hence, I'm not convinced by the conclusions that follow from this analysis. Many other studies point at the large influence of local winds on sea level variability here. It may be worthwhile to investigate the wind effects using a barotropic ocean model. These models can explain a large fraction of the observed sea level variability of monthly-mean data, as shown by Xu et al. (2015).*

In our study, Figure 18 shows the expected transport based on the assumption that wind-driven sea current is due to only a geostrophic wind approximation and a complete Ekman layer, which assumes that bathymetry does not interrupt the Ekman Layer.

However, knowing that transition area between the North Sea and the Baltic Sea is shallow, it is likely that bathymetry will interrupt the Ekman Layer, and that the water transport is more parallel to the geostrophic wind flow implied by the BANOS pattern.

We will update the text in order to clarify our assumptions about the Ekman Layer.

*8. In a paper by Chen et al [2014], the role of barotropic and baroclinic responses to the NAO in the North Sea are extensively discussed. One of their main arguments is that local density effects on a shallow shelf are small, but a horizontal pressure gradient that develops when a deep ocean column expands results in mass transport towards the shelf. How could we combine these results with your attribution process, which relies quite heavily on density effects (freshwater flux/heating)? Are in-situ temperature and salinity profiles available in the region to verify whether local density effects play a substantial role? Otherwise, regional ocean reanalysis products (some are available at http://marine.copernicus.eu) may provide estimates. From the observation of the anti-correlation between BANOS/NAO and open-ocean sea level, couldn't it be a wind-driven mass redistribution process? Over the last 15 years, you may have a look at what GRACE observations say about mass changes.*

This is an extended comment that addresses several points. The reviewer is right that the expansion of the open ocean water column may affect coastal sea-level. This effect has been estimated for the North Sea in the context of future sea-level rise due to climate change by Grinsted et al., who estimate its possible contribution with about 10% of the total sea-level rise. It may be therefore not negligible but not totally significant. We will discuss this possible contribution in the revised version by looking at the heat flux associated with the BANOS patterns.

Regarding the link between the NAO and sea-level variations in the North Atlantic (open-ocean), we feel that this is actually beyond the scope of our study that is restricted to the shelf seas.

*In-line comments*
*Note that the page numbering re-starts every page. I use the PxLy notation, referring to page x*
*line y.*

*Title and P1L6: This study mainly deals with the Baltic Sea, and only partially with the North Sea. I*
*suggest: 'German Bight' instead of North Sea.*

Here, we analysed the relation between atmospheric circulation and satellite SLAs including the whole North Sea. However, the results of our analysis show that German Bight is the most sensitive area in the North Sea to the BANOS mode of the atmospheric circulation. Therefore, the atmospheric mode that we identified mainly explains sea-level variability in the whole Baltic Sea and a part of the North Sea. At this point, we should mention that we considered the off-shore sea-level variability in the whole North Sea, but, only a part of the North Sea sea-level variability can be explained by BANOS mode of atmospheric circulation.

In addition, to quantify the contributing factors to the linkage between BANOS-index and sea-level variability, we made basin wide analysis in this region including the whole North Sea basin.

For those reasons, we prefer to keep it as 'North Sea'.

*P4L15: References to Dangendorf et al. 2013/2014 should be discussed here, and further on, what*
*do we learn from this paper that we do not know yet after reading these papers?*

We will discuss the Dangendorf et al. studies. As explained in the previous comments, there are indeed some differences of interpretation of the physical mechanism and also in the geographical areas considered.

*P5L3: For completeness, it's a good idea to add links to the web sites from which you've obtained*
*the data.*

We will add those links to web sites.

*P5L11: Do you derive season-means from monthly data? Or monthly data only over this period?*
*What about spring and autumn?*

We calculated seasonal means from the monthly data sets. The focus was on winter and summer seasons when the atmospheric anomalies are expected to be in the largest and the smallest phases.

*P5L13: Altimetry data does not have a ¼ by ¼ degree resolution: along-track observations are*
*interpolated onto a grid which can have a higher resolution than the data from which it is*
*composed. Note that observations are integrated over distances of about 100 km (See Le Traon et al*
*2001 or Pujol et al. 2014). Furthermore, observations deteriorate quickly close to land, and*
*shallow-water tides may alias into lower frequencies. Hence, it it very tricky to separate smallscale*
*features in shallow shelf seas. Tide gauges are generally more reliable in such areas. An alternative*
*may be to use along-track altimetry observations, which do not suffer from problems related to*
*interpolation. These are widely available from AVISO.*

The reviewer is right that caution is needed when using satellite altimetry near the coast, but this is the reason why we also included a comparison between the tide-gauge records and the co-located altimetry pixels. We will also better explain the spatial resolution of the satellite altimetry data sets.

*P6L2: The word 'slope' here seems a bit misleading: you compute the regression coefficient between the atmospheric pattern and sea level. 'Slope' suggests a linear trend to me.*

We do not use word *"slope"* in the mentioned line.

However, in P7L2 we had written *"The slope of the regression line is denoted as the sensitivity"*. We changed it as *"The linear regression parameter of the regression analysis is denoted as the sensitivity"*.

As well as P13L9 was *"is estimated from the slope of the regression line resulting from the regression analysis where the BANOS index"*.

We changed it as *"is estimated from the linear regression parameter of the linear regression where the BANOS-index…"*

*P6L11: The NCEP/NCAR reanalysis 1 is not really state-of-the-art anymore. Furthermore, since you use this data set to derive heat fluxes and precipitation over sea, how good does this model perform for these quantities? I guess that this model does not directly assimilate heat flux and precipitation data, and that they are derived from wind and pressure data. It may be worthwhile to use something like MERRA or ERA-interim, in which flux observations derived from satellites are assimilated. An other alternative may be OAflux (http://oaflux.whoi.edu/)*

We thank reviewer for this suggestion.

As a simple assessment test, we investigated the NCEP/NCAR net heat flux performance with respect to OAflux net heat flux. For this test, we considered field mean of a geographical area over the North Atlantic (a box covering 35° W - 15° W and 50° N-60° N) in the winter season(DJF) for the period 1984-2009. The correlation analysis on those time series indicates that NCEP/NCAR data set performs well. The correlation coefficient is 0.98.

Considering the MERRA and ERA-interim, the data sets are available from 1979 on, which is quite short in comparison to the NCEP/NCAR reanalysis data set. In this study, we analysed the contribution of atmospheric factors to the connection between BANOS mode of atmospheric circulation and sea-level. Therefore, once we established a statistical linkage between BANOS-index and sea-level variability, we could compute the strength of relation between atmospheric factors and BANOS-index by using the climatic variable as long as possible over the last century. Therefore, analysis period was not limited to satellite era. However, we can use products of MERRA and ERA-interim for the direct comparison between satellite SLAs and climatic variables in a future work.

*P6L34: This statement seems easy to verify: what is the correlation over the common altimetry/TG period?*

Probably there is a typo here - page 6 has only 28 lines. We could also not guess what could be the statement from the comment.

*P7L10: Why only check for these three? If you use all available tide gauges in the region with a substantial amount of observations, you can generate a map with the correlations at each TG location. This will make much clearer whether altimetry observations do a good job, especially at the narrow straits (Kattegat/Skagerrak etc) and around islands.*

Indeed, Figure 2 and Figure 3 provide information about the coherency of the satellite SLAs over this region as well. Since nine tide gauges have strong correlations to satellite SLAs on and around the closest point of their positions.

However, we will compute additional correlation values between tide gauges and satellite altimetry over those areas.

*P7L11: Could you make the followed procedure more clear? I don't fully understand how the data has been treated. I also wonder how you treated the seasonal cycle.*

We will clarify the text based on the suggestion. Also please see our response to point 4.

*P7L16: A correlation only does not show that signals are coherent: what about the fraction of explained variance, or a simple plot, in which both time series are compared?*
The reviewer probably means the variances themselves, since the fraction of explained variances are just the correlation squared.

The explained variances are:

|  | Winter | Summer |
|---|---|---|
| Ratan-SLA | 0.90 | 0.83 |
| Stockholm-SLA | 0.94 | 0.85 |
| Wismar-SLA | 0.78 | 0.61 |

In the following figure, we show the detrended time series of tide gauges and satellite altimetry observations, which are from the closest points to each tide gauge. In each panel, the top figure is for Stockholm, middle is for Ratan and bottom is for Warnemünde, also left(right) column represents winter(summer) values.

It can be seen from the figures that, in general, the stations agree well with the satellite SLAs in wintertime, only the Warnemünde station has some discrepancies over the beginning and end of the considered period. In comparison to the wintertime, a relative discrepancy is found between all stations and satellite SLAs in summertime, especially in the relation between Warnemünde and satellite SLAs that discrepancy becomes clearer with respect to the wintertime. This discrepancy

between satellite SLA and tide gauge in Warnemünde may occur due to the complex structure of the coast.

[Figure]

P7L17: *Are you sure that you compare the same signals with Yan et al [2004] ? Same treatment of seasonal cycle/lowpass filters/detrending etc? Maybe this statement falls outside the scope of this manuscript.*

In our study, we computed the winter means from monthly data sets, then detrended the data over the considered period prior to the correlation analysis. Since we analyse data sets in winter and summer separately on interannual time scale, we did not need to remove the seasonal cycle.

As far as we understand, Yan et al. 2004 also made correlation analysis with detrended time series based on interannual time scale (in Table II). Therefore, we think that a comparison between our results (our Table I) and Yan et al. 2004 (their Table II) is possible concerning those correlation coefficients.

P7L24: *From this correlation pattern I'd assume that the tide gauge records have a high mutual correlation. Could you show the time series of all tide gauges in one plot to verify this?*

The following figure shows the time series of associated tide gauges over the period 1993-2013. (time series represent detrended winter means (DJF) of tide gauges)

[Figure]

*P7L28: This is an interesting finding and may give some hints about the underlying processes!*

The seesaw pattern that suggests negative and positive dipole relation between North Atlantic and Baltic Sea basin wide sea-level variability also brought us to test the contribution of inverse barometer effect (IBE) to the linkage between sea-level and BANOS-mode of the atmospheric circulation. As a result, we found that major driver of the connection between sea-level and the BANOS-index was IBE.

*P8L28: How do you define 'significant'?*

We will add the significance threshold of the correlation coefficient for that record length.

*P9L3: Add here very explicitly about which time scales you are discussing.*

We will specify the time scale here, and change the sentence: *"As pointed out, the patterns of correlation between the tide gauges and the satellite altimetry fields indicate that the variations of sea-level in these regions are spatially quite uniform in both seasons (Figure 2 and Figure 3) on the interannual time scale"*

*P9L7: 'tend to be spatially coherent': again, the large amplitude variations over the region do not support this statement.*

We will remove this statement, but we would like to point out that the term *"coherent"* is sometimes used instead of the term *"strong correlation"* and that it does not necessarily provide information about the amount of the relative variation of two variables.

*P9L8: 'the influence of NAO is spatially quite heterogeneous': where do you show this?*

It has been documented in the literature (i.e. Yan et al. 2004, Hünicke and Zorita 2006, BACC Book

II 2015) and we also show in Figure 5 and Figure 6. We make clarification in the manuscript about this statement.

*P9L26: These findings contradict with the conclusion that Baltic sea level is coherent, as the difference in the found correlations is rather large, which would not be the case if the signals at both tide gauges were coherent.*

Here, we wanted to exactly highlight this point. Whereas the sea-level variability seems spatially coherent, in the sense that the tide-gauges are relatively highly correlated, the correlations with the NAO are not. This implies that there may be another large-scale effect rather than the NAO which is driving the sea-level variability over the whole Baltic Sea basin. That rationale was also the background to look for another atmospheric pattern that is strongly connected to the Baltic Sea level variability and that differs from the NAO pattern.

*P10L19: The correlation patterns seem to be almost the same as already found by Dangendorf et al, 2014a. Following the argumentation, this makes sense, as sea level variability in the Southeastern North Sea, used by Dangendorf et al, and the Baltic sea is coherent and thus has a common driver. Can't we just suffice by saying: "The atmospheric proxy, developed by Dangendorf et al. [2014] does not only work as a proxy for the S.E. North Sea, but also for the Baltic Sea."? Or is the new BANOS index doing significantly better? The only difference between Dangendorf et al and BANOS seems to me the Eastward shift in summer for the BANOS index. However the summer correlations are substantially less convincing than in winter, as I observe in figure 10. Especially the right panel in fig. 10. How does the model perform when you stick to the winter definition, even for summer?*

To statistically test the performance of the winter BANOS-index for the summer sea-level variability, we computed the 21-year running correlation values between winter BANOS and summer stations(Stockholm and Warnemünde).

[Figure]

Based on the result of this analysis, we can say that winter BANOS does not perform well to explain the summer sea-level variability. For the rest of this comment, please see our response to

point 1.

*P11L11: It seems that there are some decadal features that are shared between NAO and BANOS. Here, a wavelet analysis, as described above, may be more insightful.*

Please see our response to point 3.

*P12L8: Isn't this negative trend just a symptom of the non-stationary correlation? Something similar happened between 1905-1935.*

The reviewer is right. The correlation between the NAO and sea-level is not stationary, and this is one of the motivations of the study - namely to find an atmospheric pattern that yields more stationary correlations. The reason why the strength of the relation between sea-level and the NAO-index has decreased from 1970 onwards is not known yet. One possible reason is that the variability of the BANOS pattern becomes stronger so that the sea-level records deviate more strongly from the NAO. But this explanation would then prompt the question of as to why does the BANOS pattern become more energetic. We feel that an explanation for this behaviour lies rather in a study of the atmospheric dynamics and lies beyond the scope of this study

*P12L9: I'd say: "No significant link between NAO and Baltic sea level in summertime"*

We will add this statement into the text.

*P12L11 and the following section: Like the NAO, you find a strong anti-correlation between North Atlantic sea level and the BANOS index. That's an interesting finding in my opinion.*

Please see our response to P7L28. We will add some comments to the manuscript.

*P13L5 and the following sections: Here you show the spatially heterogeneous senitivity, again pointing at a spatially varying sea level signal. Which fraction of the variability is explained?*

Please see our response to point 2.

*P13L9: Avoid the word 'slope' here. Maybe insert a short equation:*
*dSLA = a\*BANOS with a in [mm/BANOS]*

Please see our response to P6L2.

*P13L12: It may well be the case that next to BANOS, more effects are at play here, that are not necessarily linear.*

It may be the case. Indeed, a quantitative separation of the different contributions to sea-level variations and the estimation of their possible non-linear interaction can only be done by numerical experiments with a realistic Baltic Sea&North Sea ocean model, which should be analyzed in future studies. In this study, we established a statistical linear connection between natural variations and

sea-level variability by assuming that established connection will stay same when we go backward or forward in time.

*P14L4: Why is that suggested? Horizontal pressure gradients will result in a sea level gradient due to the IB effect, and generate geostrophic winds. Do you mean that wind effects play a role?*

Please see our response to point 6.

*P14L18: As stated in the introduction: The IBE effect is the static response to pressure effects with a sensitivity of 10 mm/hPa. Since, static equilibrium is generally reached on timescales in the order of days, deviations from this static effect imply that some other effects are at play here. That's not so strange, as close to coastal areas, winds play a large role. To separate these effects, a barotropic ocean model can bring more clarity.*

We tried to explain our approach in our response to point 6. However, it is obvious that we have to clarify how we handle with the inverse barometer effect (IBE) in the manuscript.

*P15 equation 1: I'd suppose that the rate of change in steric sea level correlates with the heat flux and not the sea level itself. i.e. $dSL/dt \sim Q\_net$ instead of $SL \sim Q\_net$. What if you integrate $Q\_net$ before computing any correlation?*

The reviewer is right, but, we actually did integrate the heat flux over the whole season, so that the variable that we use to correlate with the seasonal sea-level is the total amount of heat that goes into the ocean (or leaves the ocean) over one season

*P15L24: This number is rather large, I suspect it's incorrect.*

This number is computed from the sensitivity of net energy flux to one unit change in the BANOS-index. Sensitivity value is 3.28 J/m².s. This means one unit increase in the BANOS-index causes 3.28 J/m² per second over the winter. To compute the sensitivity of net energy flux to the BANOS-index over one winter, we multiplied the 3.28 with 60*60*24*90, which is equal to that number. That amount of heat distributed over the upper 50 meters in the water column would rise the water temperature by about 0.1 K, so it is not clear to us what the reviewer means. We will double check this number.

*P15L25 and below: How did you compute this? If I'd compute the thermal heating that result of the afore mentioned number, the whole Baltic sea would evaporate rather quickly. Are there insitu T/S observations, or SST observations that can confirm the large impact of density changes induced by local heat fluxes? I'd guess that on a shallow shelf, the effect of density changes is rather limited. Furthermore, if so much water evaporates or rains into the basin, doesn't the resulting sea level change lead to transport with the open ocean?*

Please see our response to P15 equation.

*P16L11 and beyond: Like with energy fluxes, isn't it expected that sea level varies according to the integral of the total freshwater flux?:*
*dSL/dt ~ E*
*Furthermore, do you suppose that the total mass in the Baltic Sea is affected, or that the effects are caused by changes in the salinity?*

We actually use the water flux integrated over the whole season.

*P16L25: The correlation pattern does not tell much about what causes what. I'd say that the precipitation/evaporation pattern changes and sea level changes are both caused by the BANOS-related pressure changes. Therefore, they show mutual correlation. But that does not show that the P-E flux causes sea level changes! Hence, the conclusions reached from P16L32 onward are not really justified in my opinion without further proofs.*

The reviewer is right that the statistical analysis is not proof of causal relationships. The statistical analysis is rather an estimation of the possible contribution of the fresh water flux and is to be understood as an analysis of plausibility.

*P17L7: Do you mean 'geostrophic wind flow' here?*

Yes, we mean geostrophic wind flow. We changed the text accordingly.

*P17L8: I don't understand what you mean here: why can't the BANOS-induced wind forcing transport surface water between both basins? Although for different regions, many studies point at the large impact of local wind variations on monthly and interannual sea level, including Sterlini et al. 2016, Dangendorf et al. 2013, 2014a, 2014b, and many more. Does the same happen in the Baltic Sea? Figure 3a in Dangendorf et al, 2014a clearly suggest a wind set-up effect.*

In wintertime, the SLP BANOS pattern attributed geostrophic wind flow does not indicate westerly winds over the transition zone. It rather implies south-westerly wind, which can cause strong water accumulation towards west Norwegian coast and German Bight. Hence, we can speculate that water accumulation (coastal downwelling) can cause water transport from German Bight and west Norwegian coast towards the Baltic Sea.

*P17L15: To my knowledge, as described above, both the local bathymetry and shallow water, as well as the presence of coasts render the Ekman transport assumption invalid. The width of the basin is about 100 – 200 km, which is probably smaller than the Rossby radius of deformation. Especially around Skagerrak and Kattagat, the basin dimensions become very small. Hence, I would not trust results based on the Ekman transport assumption. Again, a wind-forced barotropic ocean model or a regional ocean reanalysis could bring more trustworthy results regarding changes in wind-induced transport and sea level variability.*

In our discussion we provide an estimation of the potential contribution of Ekman pumping under 'perfect' conditions. We are aware that these conditions are not totally met, even less so in the shallow straights connecting the North and the Baltic Sea. However, our estimation is not focused on these areas. We wanted to estimate the transport that can be attributed to the large-scale BANOS

pattern. In the end, the total water transport will be caused by a combination of the local wind forcing and the larger scale transport, which will also cause local pressure gradients along the narrow straights.

The reviewer is right that a quantitative estimation requires the use of a comprehensive ocean model, but our study is a statistical analysis of the available data.

*P18L3: The impact of NAO/BANOS-related variability is the only atmospheric effect on sea level analysed in this study. There may be more atmospheric processes affecting sea level on interannual time scales.*

We agree with the reviewer's comment. It is also reason that we write "partly quantified".

*I'm afraid that most conclusions are not justified by the presented results:*
*• Conclusion #1: According to other studies, the amplitude of the interannual variability differs widely around the region and therefore, the variability is not coherent. Furthermore, in figure 14 you show that the spatial signal is far from coherent, as the sensitivity values differ by a factor 10 over the basin.*

The comment by the reviewer is based on a different meaning of coherency. The reviewer uses this term when the records would show roughly the same amplitude of variability, whereas we use it to denote correlated behaviour in time. We will clarify this point in the revised version.

*• Conclusion #2: In figure 6, top I see a rather strong correlation between the NAO and the altimetry-derived sea level in wintertime, rather than weak!*

We derive this conclusion from the 21-year running correlation values between the NAO and two tide gauges over the last century. The temporal variability of the link between the NAO and sea-level variability can sometimes show some strong correlation for an individual short term period, as it is happened over the satellite era.

*• Conclusion #3: There are essentially two indices: summer-BANOS and winter-BANOS.*

We will change this accordingly.

*• Conclusion #4: The BANOS index only correlates with sea level variability in the German Bight, and not in the whole North Sea in summer.*

Our conclusion #4 does not say something inconsistent to this comment.

*• Conclusion #5: Since the regression coefficient deviates from the static IB response, it's probably not only the IB effect that causes the pressure-sea level link. Furthermore, if the IB effect explains 88 % of sea level variability, and surface fluxes 35%, we explain more than 100 %. Again, you've only showed a correlation pattern and not what causes what. They also may have a common cause. How are these percentages derived? The conclusions regarding wind-driven variability depend on the Ekman transport approximation, which is probably not valid in this region.*

One of the complexities in identifying the physical mechanism of the sea-level variation is that there can be interrelations among the considered physical mechanisms. Hence, a possible overestimation may occur if some interdependent forcings (predictors) are included in the associated statistical model. This possibility can be partially excluded with the use of atmosphere indices that contain several effects and represent them in a single index (e.g. Sterlini et al. 2016). For example, the NAO, as a mode of atmospheric variability, can carry information about the wind, sea-level-pressure and surface heat fluxes. In this part of the study, describing relations between the effects of the related driving factors and sea-level variation based on atmospheric indices enables us to use those relations without making any further analysis such as multicollinearity test between drivers. Considering the explained variances of sea-level by the inverse barometer effect (IBE) and net energy flux (NEF), it seems that sea-level variance is overestimated due to the amount of explained variance in total. The first reason is that we assumed a complete equilibrium over the Baltic Sea and North Sea region for the IBE, which is in real not the case. Broadly speaking, we estimate a possible maximum contribution of the IBE to the BANOS attributed sea-level variability over the study area. The second reason is that the impact of the NEF is computed by taking the spatial average of the Baltic Sea and North Sea basins. The amount of thermal expansion of the water per one unit change in the BANOS-index would differ depending on the assumed average value of temperature, salinity and pressure through the water column (for this, please see our response to P15 equation).

Here, it should also be noted that a high correlation does not necessarily mean a strong direct physical connection between the conducted factor and sea-level variability. Therefore, the statistical analysis that we applied in this study investigates the potential of contributions of the considered physical factors to the sea-level variability. The quantitative attribution of the driving factors to the sea-level can only be described by numerical experiments with a realistic Baltic Sea and/or North Sea ocean model (e.g. Kauker and Meier 2003).

*Figures,*
*1. In general, it may be a good idea to avoid the 'rainbow' color scale for correlations. A good summary of which color maps are suitable can be found here:*

*https://betterfigures.org/2015/06/23/picking-a-colour-scale-for-scientific-graphics/ I'd suggest to use a 'diverging' color scale. It looks like you use GMT for the plots, for which many good diverging color palettes can be found here:*
*http://soliton.vm.bytemark.co.uk/pub/cpt-city/jjg/cbcont/div/index.html*

We used "MATLAB" software for all plots. We will try to enhance the colour scale.

*2. It may also be a good idea to contour areas with significant correlations*

A contour delineating the significant correlations will be added to the figures

*3. Some figure captions can be expanded to describe the followed procedure. For example figure 4: "The correlation pattern between de-trended sea level during the winter months(DJFM) and the detrended NAO index over the same months. The correlation has been computed between January 1993 and December 2014" or something similar. This will allow easier reproduction of your results.*

We will add expanded captions to the figures.

*Figure 1: Maybe add the locations of the tide gauges*

We will add the locations of the tide gauges.

*Figure 10: On the left, some data seems to be missing*

To compute the correlation coefficients, we prescribed a threshold of 75% data availability for the considered period.

REFERENCES (in addition to reviewer's references)

Andersson 2002, *Influence of Long-Term Regional and Large-Scale Atmospheric Circulation on the Baltic Sea Level*. Tellus A 54 (1): 76–88.

Stramska, M. 2013. *"Temporal Variability of the Baltic Sea Level Based on Satellite Observations."* Estuarine Coastal Shelf Science 133: 244–50.

Kauker and Meier 2003, *Modeling Decadal Variability of the Baltic Sea: . Reconstructing Atmospheric Surface Data for the Period 1902– 1998*. Journal of Geophysical Research 108 (C8): 3267.

Hurrell, J. , Y. Kushnir, G.r Otterson, and M. Visbec. 2003 . *"An Overview of the North Atlantic Oscillation."* The North Atlantic Oscillation: Climatic Significance and Environmental Impact 134: 263. doi:10.1029/GM134.

Hünicke and Zorita 2006, *Influence of Temperature and Precipitation on Decadal Baltic Sea Level Variations in the 20th Century*. Tellus A 58 (1): 141–53.

Cattiaux and Cassou 2013, *Opposite CMIP3/CMIP5 trends in the wintertime Northern Annular Mode explained by combined local sea ice and remote tropical influences*, Geophysical Research Letters, vol. 40, 3682-3687.

BACC Book II 2015, *Second Assessment of Climate Change for the Baltic Sea Basin*. Edited by BACC II Author 15 Team. Springer.

---

## Author Comment (AC2) · 13 Jun 2017

We thank the reviewer very much for reviewing our manuscript, for providing constructive criticism and useful suggestions. We respond to all comments below.

*Specific Comments*
*A major part of the paper is the presentation and discussion of how much better the BANOS-index correlates to sea level changes in the Baltic Sea than the NAO-index. It would be helpful to the reader to discuss the BANOS-index with respect to other slp indices that have been used for the Baltic Sea. For example the BAC index in Andersson, 2002 or the BSI in Lehmann et al, 2002. How much different are those indices from the BANOS-index? Would the slp gradient over the transition zone between North Sea and Baltic Sea be different between BANOS and BAC or BANOS and BSI?*

To answer the questions raised by the Reviewer, we initially contacted Andersson in order to obtain the time series of the BAC-index, but unfortunately we were informed that time series is not available any more. Therefore, we cannot make a statistical comparison between the BANOS-index and the BAC-index. Also, the BAC-index is constructed with a different method than the other indices (BSI-index, NAO-index and BANOS-index).

We also contacted Lehmann, he kindly shared the time series of the BSI-index with us. The BSI-index time series starts in 1948. Therefore, we made a correlation analysis between the BANOS-index and the BSI-index for the period 1948-2013. The time series are detrended prior to the correlation analysis. The results are shown in the following table.

| Corr. Coef. (period 1948-2013) | Winter | Summer |
|---|---|---|
| BANOS - BSI | 0.91 | 0.64 |
| NAO- BSI | 0.72 | 0.19 |
| BANOS-NAO(corr. in the manuscript) | 0.68 | -0.12 |

Briefly, this correlation analysis indicates that the indices, BANOS and BSI, may share some similarities in wintertime, but the strength of the relation between those indices becomes much weaker in summertime. Moreover, the correlation between NAO and BSI indices seems very weak in summer, although it is statistically significant. On the other hand, there is neither a significant nor a positive relation between NAO and BANOS indices detected in summertime.

We should additionally mention that Lehmann et al. (2002) constructed the BSI index only from the analysis of winter months. They defined the BSI-index as the difference of normalised SLP anomalies at the positions of (53°N30', 14°E30') and (59°N30', 10°E30') by considering the period 1979-1998. This shows that they selected different SLP fields than we selected for the winter BANOS-index. Moreover, they concluded that a positive BSI pattern is linked to westerly winds over the Skagerrak and Kattegat, whereas in the winter BANOS-index the orientation of the winds is rather north-westerly. Regarding the BANOS-pattern related sea-level variability; we interpreted a large impact of the inverse barometer effect (IBE), which transports water from the North Atlantic towards the North Sea and the Baltic Sea. In addition, Lehmann et al. (2002) is more focused on monthly and weekly variation of the Baltic Sea level variability. Our study analyses sea-level

variability on the interannual time scale.

We should also note that similar question was raised by the Reviewer 1. The reviewer suggested a comparison between Dangendorf et al. (2014) * atmospheric proxy and the BANOS-index. Please also see our response to the first point of Reviewer 1's revisions.

*Dangendrof et al. (2014), A new atmospheric proxy for sea level variability in the southeastern North Sea: observations and future ensemble projections, Climate Dynamics, 43:447.

*Is the BANOS-index usable without a gridded slp field? Can it be inferred from station data (e.g. Stockholm - Odessa), like the NAO-index and would it show good correlation the one derived from gridded data?*

The station-based NAO-index is computed from the differences between two normalized SLP fields. Here, we used the same method. What the reviewer asks is whether it is possible to construct a BANOS-index from station data. It could be possible, but individual station pressure records can be affected by local effects - small-scale and transient meteorological factors - unrelated to the large-scale mode of atmospheric circulation. This problem is also addressed by Hurrell et al. via:

https://climatedataguide.ucar.edu/climate-data/hurrell-north-atlantic-oscillation-nao-index-station-based

Since we have the gridded SLP fields for the period of analysis, we prefer to use those data that have also been profusely used in other studies.

*How sensitive are the results and the conclusions for summer when the summer was defined as the months JAS? Baltic Sea sea level in summer show little variation during these three months (e.g. Hünicke and Zorita, 2008, Meier et al., 2004). The month of June usually lies between the spring minimum and the summer values. What is the reason to choose June, July and August as the summer season?*

Some authors have used other months to compute summer means. The idea by selecting those months (JJA) as summer months were that the smallest variance of sea-level occurs in those months over the last two centuries (i.e. Andersson , 2002- Figure 3).

To measure the robustness of the BANOS-index based on the method suggested by the Reviewer, we computed the 21-year running correlations of JAS mean summer variations between BANOS-index and sea-level. We also include the 21-year running correlations between BANOS-index and sea-level for the JJA summer season. Results are shown in the following figure.

[Figure]

The figure indicates that the Stockholm sea-level variability can be well represented by the BANOS-index for the summer of the JAS months. For the JAS summer, those correlation values - for BANOS-index and Stockholm sea-level- indicate a steady relation over time and are always higher than 0.65 over the 20st century. We should note that the relation between BANOS-index and Stockholm for the JAS summer is even stronger and more stable in time than the relation between the BANOS-index and Stockholm for the JJA summer. However, Warnemünde is weakly connected to the BANOS-index for the JAS summer over the last century.

*The title of the paper suggests the investigation of North Sea and Baltic Sea sea levels. In the derivation and evaluation of the BANOS-index only Baltic Sea sea levels are taken into account. From Figures 4 and 13 one would expect that the correlation of winter mean sea levels in the North Sea improve over the NAO correlations. For summer at least in the eastern part of the North Sea. Please consider to show one or two tide gauges from the North Sea in Figure 12, or add a figure like Figure 12 for two North Sea tide gauges.*

We will add tide gauges along the North Sea coasts.

Please, also see the following table for the correlation coefficients among the BANOS-index, the Cuxhaven (settled on the coast of German Bight) and Stockholm stations for the period 1900-2008.

| Corr. Coeff. Winter (Summer) | Cuxhaven | Stockholm |
|---|---|---|
| BANOS | 0.80(0.50) | 0.84(0.72) |
| Cuxhaven | - | 0.88 (0.67) |

*In the abstract it is said that the wind associated with the slp pattern of the BANOS-index does not lead to transport of water into the Baltic Sea. The analysis in Section 4.8.4 is a good idea, but since the return flows along the coasts and at depth are not taken into account, the statement might not hold up to a more thorough investigation.*

In the abstract, we consider the case that the geostrophic wind linked to the SLP BANOS pattern does not suggest a westerly wind over the transition zone between the North Sea and Baltic Sea in wintertime. Therefore, the geostrophic wind would not push surface water from the North Sea into the Baltic Sea over the transition zone, where the bathymetry is expected to interrupt Ekman Layer. It rather plays a role in accumulating water (downwelling) towards the west coast of Norwegian Trench and German Bight.

However, the geostrophic wind linked to the SLP BANOS pattern suggests a possible transport of surface water from the North Sea into the Baltic Sea in summertime, since BANOS related geostophic wind is expected to be westerly over the transition zone, as in the case of the NAO pattern.

In our study, we did not consider the return of the water flow.

*I would like to see the correlation coefficients for all of the nine stations in Table 1. Future studies might benefit from that information and for the present study the information might help with the interpretation of Figures 2 and 3.*

We will include the correlation coefficients of all stations in Table 1.

*Good correlations between altimetry and tide gauge data is said to indicate progress in satellite altimetry. Could the source of the improvement over earlier comparisons be specified? It is probably not the altimeter instruments themselves that have progressed so much. Information on the geoid? New algorithms? Amount of data?*

Primarily, updating the geophysical corrections and usage of refined mapping parameters should be the reasons improving the correlation between altimetry and tide gauge data.

We should also indicate that main improvement of DT2014 SLA data set is achieved by changing the reference period of the SLA products to a new altimeter reference period. This reference period takes the advantage of the 20 years observations and optimizes the reduction of along-track random noise, which was largely involved in the physical signal of the previous version DT2010. For the further information, we suggest the paper of Pujol et al. (2016) - http://www.ocean-sci.net/12/1067/2016/os-12-1067-2016.pdf .

The core of improvement from their study: *"Numerous innovative changes have been introduced at each step of an extensively revised data processing protocol. The use of a new 20-year altimeter reference period in place of the previous 7-year reference significantly changes the sea level anomaly (SLA) patterns and thus has a strong user impact. The use of up-to-date altimeter standards and geophysical corrections, reduced smoothing of the along-track data, and refined*

*mapping parameters, including spatial and temporal correlation-scale refinement and measurement errors, all contribute to an improved high-quality DT2014 SLA data set"*

*In Section 4.8.1 yet another index is introduced as the slp differences between two geographical locations that differ from the ones introduced in Section 4.4. Would you please explain why there is a need to introduce different locations where to measure slp to get at the IBE contribution? Why Denmark Strait and not Labrador Sea for the summer for example? But more importantly why not stick to the definition (5W, 45N) - (20E, 70N) for winter and (30E, 45N) - (20E, 60N) for summer?*

We needed geographical points where sea-level data were available in order to estimate the sensitivity of BANOS attributed sea-level to the IBE. Therefore, the points that we selected were the optimum ones according to winter-summer BANOS patterns and sea-level data availability.

*In Section 4.8.2 please indicate in the text whether the whole water column was heated or just the mixed layer (down to what depth?) to absorb the energy.*

The estimated explained variance (35%) shows the potential of net energy flux contribution to explain the linkage between the BANOS-index and sea-level variability. Here, we assume that the thermal expansion coefficients do not vary with temperature or pressure. It is, therefore, a first order estimation.

Under those assumptions, we estimate that 1 unit increase in the BANOS-index can cause 1 mm sea-level rise due to the contribution of net energy flux. This estimation is independent of water depth under the stated assumption that the thermal expansion coefficients are temperature and pressure independent.

*Section 4.8.4 discusses an interesting point but the argumentation stops halfway through. How is the Ekman transport different in the transition zone between North Sea and Baltic Sea for NAO- or BANOS-index related patterns? And what are the consequences for the sea-level in the Baltic Sea?*

We will clarify the text according to suggestions.

*In Section 4.8.4 it is argued that during summer BANOS and NAO related wind forcing (slp gradients) could be similar. I thought the main reason to introduce the BANOS index was the liberty to define an index separately for winter and summer, so that missing correlation for summer sea-level variability in the Baltic Sea could be explained. This section would benefit from a rewrite, I suppose.*

We will rewrite the text for the clarification.

*The Conclusions (page 18, line 28-29) mention that there is no contribution of NEF in summer. Section 4.8.2 states that summer has not been included in the analysis, because of negative correlations. Please reformulate the sentence in the conclusions, page 18, line 28-29.*

We will reformulate the sentence.

*For the last part of the Conclusions (page 19, line 2-4) the evidence is missing. See also the comments above on Section 4.8.4 and the Abstract.*

We will clearly write the evidences in the related parts of the manuscript.

*The conclusion might benefit from a restructuring. The itemized list could be shortened to contain the keywords only as a summary of a continuous text around it.*

We will update the conclusion.

*Technical Corrections*
*page 1, line 7: interannual time scales.*
We will change this accordingly.

*page 2, line 24: NAO-index describes weaker*
We will change this accordingly.

*page 3, line 19: Andersson (2002) who focused on*
We will change this accordingly.

*page 3, line 20: Dangendorf et al. (2014) who investigated the*
We will change this accordingly.

*page 3, line 20-21: investigated the North Sea, reported that atmospheric variability that differs from the NAO may still explain part of the sea-level variability.*
We will change this accordingly.

*page 3, line 32: different sorts of data sets*
We will change this accordingly.

*page 4, line 32: This study focuses on the winter*
We will change this accordingly.

*page 5, line 1: this study and the following section*
We will change this accordingly.

*page 5, line 5-7: Although the threshold of any computation involving tide gauge records was set at 75% availability of data for the considered period, seasonal means are calculated in case of availability of tide gauge records for two months. [I am not sure whether I understand it the way the authors mean it. Could this information be written a little more clearly?]*

We will clarify the text.

Here we would like to mention that for the analysis period like 21-year running correlations, we prescribed a threshold of 75% data availability. However, we computed seasonal means if two months were available (means 66.6% availability of data set to compute seasonal mean).

*page 5, line 23: the northeastern boundary of [If Smögen is meant by the northeastern boundary of the North Sea I recommend to call it Skagerrak.]*
Here we write the names of tide gauges. When we mention about the region we write Skagerrak (i.e. Figure 1).

*page 6, line 14: and produces a complete gridded data set*

We will change this accordingly.

*page 6, line 16: resolution of a T62 Gaussian grid [Or mention the resolution in deg or km]*

Reanalysis data set has no regular grid. We will change it as the following.

Old version : "*This data set has a spatial resolution of 192x94 points with T62 Gaussian grid covering the Earth's surface*"
New version: "*This data set has spatial resolution of a T62 Gaussian grid with non-regular 192x94 points covering  the Earth's surface*"

*page 7, line 9: examine the correlation of the satellite [see below]*

Old version:"*We first examine the coherence of the satellite altimetry observations with the tide gauge records.*"
New version: "*We first examine how the satellite altimetry observations covary with the tide gauge records.*"

*page 7, line 11-13: The seasonal means ... linear regression. [This has been said in the methods section. It could be scratched.]*
We will remove that part from the text.

*page 8, line 30, 32 in the southern Baltic Sea*
We will change this accordingly.

*page 9, line 5: Figure 4-lower [see also comment below]*
We wil change this accordingly.

*page 9, line 28: does seem to be strongly connected to the NAO-index from 1998 in wintertime. OR does not seem to be strongly connected to the NAO-index until 1998 in wintertime.*

We will correct the text. Here, the correct version should be: *"does not seem to be strongly connected to the NAO-index until 1998 in wintertime"*

*page 9, line 30: could indicate the existence of*
We will change this accordingly.

*page 10, line 30: from what the NAO implies.*
We will change this accordingly.

*page 11, line 33: most of the time the gliding*
We will change this accordingly.

*page 13, line 31: region on the interannual time scale. OR region on interannual time scales.*

We will correct the text. It will be "*on the interannual time scale*"

*page 14, line 27: (3.44\*18.09) OR (3.44 ncdot 18.09) AND (1.39\*7.02)*
We will change this accordingly.

*page 15, line 3: radiation reaching the earth surface and longwave (LW) emitted*
We will change this accordingly.

*page 15, line 9: [SH and LH do not correspond to the naming in the equation.]*
We will change this accordingly.

*page 15, line 18: estimated as 3.28 W m-2 u-1)*
We will change this accordingly.

*page 15, line 24: of 25,505,280 (W m-2) per one unit [How much change is that? It might be easier to grasp by using something like 2.5 10 ˆ7 (W m-2)]*

We will modify the numerical expression by using the suggestion of reviewer. :

2.5e7 (J m-2 yr-1)

For the technical question, please see our response to previous question: "In Section 4.8.2 please

indicate in the text whether the whole water column was heated or just the mixed layer (down to

what depth?) to absorb the energy. "

*page 15, line 24: of 25,505,280 (W m-2) per one unit [Should the units be (J/winterm-2) instead of (J/s m-2)?]*

We will write it as 2.5e7 (J/m-2 yr-1)

*page 16, line 9: (2257 kJ kg-1)*
We will change this accordingly.

*page 16, line 16-17: [During summer the western North Sea is not similar in precipitation and freshwater flux.]*

Old version: *"In summertime, the correlation patterns between the BANOS-index and precipitation*

*and the between the BANOS-index and freshwater flux display similar results"*

New version: *"In summertime, the correlation patterns between the BANOS-index and precipitation*

*and the between the BANOS-index and freshwater flux display similar results other than the western*

*part of the North Sea"*

*page 16, line 29: [It would be more concise to use the regions defined in Figure 1 when describing the correlation pattern.]*
We will update the text based on the suggestion.

*page 17, line 2: of sea-level would reach  10 mm per one unit*
We will change this accordingly.

*page 17, line 3: [Either remove the last sentence or explain why the high correlation in the north eastern part of the North Sea drainage basin (incl. Norway) does not contribute to North Sea sea level variability.]*
We will remove the last sentence.

*page 17, line 14: be similar to the case of the NAO.*
We will change this accordingly.

*page 17, line 15: assuming that the Ekman layer*
We will change this accordingly.

*page 17, line 16: interrupted by bathymetry.*
We will change this accordingly.

*page 17, line 21: if bathymetry would not interrupt*
We will change this accordingly.

*page 17, line 22: is generated by the BANOS-related north-easterly winds*
We will change this accordingly.

*page 18. line 1: [The Norwegian coastline is in the north-eastward direction. That contradicts the previous argumentation.]*
Old version: *"...towards the Norwegian, German, Dutch and UK coastlines"*
New version: *"...towards the German, Dutch and UK coastlines"*

*page 18, line 4: on interannual time scales. OR on the interannual time scale.*
We will change it as *"on the interannual time scale"*

*page 18, line 15: is more important for the*
We will change this accordingly.

*page 18, line 32: [The last sentence seems to contradict what has been said in the first sentence (line 30-32). Would you please specify which contributions of the freshwater flux are negligible.*
Old version: *"... is negligible in the Baltic Sea and the North Sea"*
New version: *"... is negligible in the Baltic Sea and the North Sea in wintertime"*

*page 19, line 4: to that related to the NAO in*
We will change this accordingly.

*caption Figure 14: Note the different intervals on the color scales.*
We will change this accordingly.

*Readability: The readability of Section 2 could be improved by omitting the sub-subsections.*
We will omit subsections in the Section 2.

*Figure captions: I'd recommend to change the naming of the positions in the plots from down to lower/bottom and from up to upper/top.*
We will change this accordingly.

*Coherency or correlation (page 7, line 9ff): Did you really look at correlations on different time scales or is "coherence" used here as a synonym for correlation? The expression coherence or coherency appears more often later on. It would be helpful to specify what is meant by coherency or perhaps just use correlation.*

In the manuscript we use the term coherency in the general sense and not in the statistically related sense. We will clarify this point in the new version.

*Station Smogen (page 5, line 24ff): It appears in some places in the text, it should be Smögen.*

Sea level data is provided by PSMSL (www.psmsl.org). In their webpage, the name of station is written as "Smogen". We will use the correct name and indicate that the name in the PSMSL is not totally correct.

*Caption Figure 5: for the winter (solid line) and summer (dotted line) seasons.*
We wil change this accordingly.

*Figure 8-9: These two figures could be combined in the same way Figure 10 was done.*
We will change this accordingly.

*Figure 10: Why is the northern half of the figure blank? If possible, the figure should be redone. It avoids unnecessary doubts.*

We will replot Figure 10 based on the suggestion of the reviewer.

To compute the correlation coefficients, we put the threshold for data availability to 75% for the considered period.

*Figures 8-10: The reader might appreciate the eye be guided with a mark on the plots, where the BANOS-index is defined.*

We will change it accordingly.

*Figure 11: It is hard to see similarities and even more so difference between the two indices. An additional running mean could improve the figures. Red-blue bar plots like they are used for the display of the NAO-index might be an option.*

We will change the plot by using the Red-blue plots, and include running means in the Figure.

*Caption Figure 18: It is not clear which index or which season is displayed in the upper and lower panel of Figure 18. Also, for the sake of completeness it would be good to indicate the units and a reference vector.*

We will partly rewrite the caption to clarify the winter and summer seasons. Since these representative vectors are constructed from correlation gradients, they are unitless.

*Sub-sub-section 4.8.2: It would be clearer to use one name only for the net energy flux. Either Qnef or NEF. Also, in the text the "net"- and "nef"-part of names like SWnet should be a subscript. Or maybe drop the "net" altogether in the whole section.*

We will use NEF instead of Qnef for the whole section.

*Sub-section 4.8: In my opinion it would suffice to indicate the sensitivities with one digit after the floating point.*
We will change this accordingly.

---

## Author Response (AR2)

We thank reviewers for their further help with the manuscript. All suggestions of the reviewers have been considered and the manuscript has been accordingly modified. Please see our point-by-point responses below.

**REVIEWER #1**

-page 2, line 15: centred over the Azores

**Answer:** The text has been changed accordingly.

-page 4, line 15: pattern and whether

**Answer:** The text has been changed accordingly.

-page 4, line 16: more stable connection in time with sea-level

**Answer:** The text has been changed accordingly.

-page 4, line 29: observations onto a 1/4°x1/4° resolution

**Answer:** The text has been changed accordingly.

-page 5, line 22: found in

 https://www.aviso.altimetry.fr/fileadmin/documents/data/duacs/Duacs2014.pdf. [Could this report be cited as a technical report by CNES/AVISO? I would make the text more readable.]

**Answer:** The text has been updated based on the suggestion.

New version*:" found in AVISO/CNES product report published in April 2014"*

-page 5, line 32f: The names of some stations listed in the PSMSL database like Smögen, Warnemünde and Travemünde are not totally correct. [I think this comment is not necessary. The discrepancy is probably due to the adoption of ASCII to represent the alphabet of other languages.]

**Answer:** This indication has been suggested by another referee during the first review. However, we agree with the current reviewer. This sentence has now been removed from the manuscript.

-page 7, line 8: each of the nine tide gauges

**Answer:** The text has been changed accordingly. We specially thank the reviewer for his/her careful reading.

-page 7, line 15: altimeter data is provided by

**Answer:** The text has been corrected accordingly.

-page 15:, line 24ff: SWnet [If possible, it would be nice to have the "net" part as an index to SW, like in equation 1.]

**Answer:** The text has been changed accordingly.

-page 18, line 24: could flow more parallel

**Answer:** The text has been corrected accordingly.

**REVIEWER #2**

I have a couple of more general questions. First, the correlation already implies a linear dependence, as does the linear regression based sensitivity analysis, what is then the value to do and present results from both approaches? This could be explained in the Methods section.

**Answer:** By estimating the sensitivity values of sea level over the offshore and coasts, we intended to reveal the relative sensitivities of sea level to the atmospheric circulation in different locations. It was sometimes the case that although they have the same correlation values, sea level sensitivity values indicated different amplitude of effect of atmospheric indices on sea level variability.

The following sentence has been added to the method section: *"It should also be noted that the estimation of sensitivity values improves the understanding of the connection between atmospheric circulation and sea level, since the sensitivity values of sea level to the atmospheric index may differ in a case that correlation values between those sea level variations and atmospheric circulation indicate the same covariation. This helps identify more vulnerable off-shore and coastal locals in responding to the atmospheric circulation."*

Second, the authors compare the BANOS-index with the Dagendorf et al. (2014) proxy. There are a number of other atmospheric circulation indices which are more local than the NAO and may therefore explain the sea-level variability with a comparable skill to the BANOS-index. Closest to the study region are the Scandinavian, the Polar/Eurasian and the East Atlantic teleconnection patterns.

As these patterns are already commonly adapted, it would be fair to see at how they co-vary with the North/Baltic Sea level.

**Answer:** As the reviewer mentions, there are several studies such as Anderson, 2002; Lehmannet al., 2002; Dangendorf et al., 2014 constructed new atmospheric index for the explanation of sea level variability in this region. To see the covariability between those two indices BANOS and Dangendorf et al.2014 proxy, we computed the correlation coefficients of de-trended time series between those two indices. For wintertime (summertime) the correlation is 0.89 (0.74) for the period 1900-2008. These values indicate that sea level variability in this region will be substantially explained by the Dangendorf et al. 2014 proxy.

Similar question was addressed by the reviewers during the first round. Please also see our first round responses to Referee 1-comment 1, Referee 2-comment 1 for detailed answer to the current comment.

Specifically for this suggestion, we analysed the strength of the relation between the representative tide gauges (Stockholm, Warnemünde and Cuxhaven stations) and three teleconnection pattern indices in wintertime and summertime for the period 1950-2013. The time series of teleconnection patterns were available through: http://www.cpc.ncep.noaa.gov/data/teledoc/telecontents.shtml.

Please see the table showing the correlation computations below (time series are detrended prior to the correlation computations and the 95% significance level is 0.24 ).

| Winter/Summer | Scandinavian | Polar Eurasian | East Atlantic |
|---|---|---|---|
| Stockholm | -0.67/-0.47 | -0.03/-0.41 | -0.18/0.28 |
| Warnemünde | 0.11/-0.27 | -0.21/-0.25 | 0.04/0.27 |
| Cuxhaven | -0.60/-0.47 | -0.10/-0.34 | -0.06/0.45 |

The correlation table indicates that the sea level variation in the Baltic Sea and the North Sea tide gauge stations mostly weakly covary with those teleconnection patterns. In wintertime, stations have mostly non-significant correlation values. Only the Scandinavian teleconnection pattern shows significant with the Stockholm and Cuxhaven tide gauges in wintertime. In summertime, the correlations are also not quite strong. The largest correlation value measured between Scandinavian pattern and the Stockholm and Cuxhaven stations is -0.47.

Accordingly, we have added the data&results of this analysis into the manuscript (For that addition, please see both the last part of the Data Sets Section and again the last part of the Subsection 4.5).

Minor comments:

- page 1, lines 11-12. 'This circulation pattern displays a circulation pattern ...' does not make sense. Just say 'This circulation pattern is more closely ...'.

**Answer:** Probably there was a mismatch in the lines encountered during the reading. The sentence reads: "This circulation pattern displays a link to sea-level that remains stable through the…". Therefore, we leave the sentence as it is.

- page 3, line 7. '...snow and ice melt...'.

**Answer:** The text has been changed accordingly.

- page 7, line 13. '...between the closest SLA grid cell...'.

**Answer:** The text has been changed accordingly.

- page 8, line 1. One line paragraphs are too short. Combine this line with the paragraph starting in line 5.

**Answer:** For fluency of the text, we now have moved that singular line to the previous paragraph.

- page 9, lines 12-19. The line of thinking is somewhat unclear here. I would focus on seasonally heterogeneous response instead of spatial one and write: '.. in contrast, especially seasonally quite heterogeneous ...'. This change would make the paragraph more logical, I think.

**Answer:** We specially thank the reviewer for this suggestion. To clarify the logic of the hypothesis formulation, we have placed the following sentence in that section: *"In addition to that, the connection between the NAO mode of atmospheric circulation and sea level variability in this region has seasonally a quite heterogeneous pattern."*

- page 9, lines 14-15. Drop the unnecessary sentence: 'We also show that heterogeneity in the following figures.'

**Answer:** The sentence has been removed from the text.

- page 9, line 18. '... North Sea, both in the winter ...'.

**Answer:** The text has been updated accordingly.

- page 9, lines 18-19. 'At this stage, we investigate which patterns ...'.

**Answer:** The text has been updated accordingly.

- page 19, line 19. '... Stockholm, Cuxhaven and Warnemunde stations ...'.

**Answer:** The text has been changed accordingly.

- page 13, line 1. '...both the BANOS-index and the NAO-index (Figures 4 and 12), ...'.

**Answer:** The text has been updated accordingly.

- page 13, line 14. 'Given not only the relatively stable correlation ...'.

**Answer:** The text has been updated accordingly.

- page 13, line 14. '...coastal sea-level variations compared to the NAO.

**Answer:** The text has been updated accordingly.

- page 13, line 22. Make this sentence the last one of the previous paragraph.

**Answer:** The text has been updated accordingly.

- page 14, line 2. I would say: 'depending on the exact location of the SLP grid cells selected to construct the BANOS-index ...'. You have already defined the BANOS-index to be based on variable SLP grid cell locations.

**Answer:** What we mean here is that the sensitivity values will differ based on the definition of the BANOS-index time series. Therefore, we would like to keep the sentence as it is.

- page 14, line 11. Just say geostrophic winds instead of horizontal gradients in air pressure. It is clearer to the reader.

**Answer:** The text has been updated accordingly.

- page 14, lines 12-14. What confuses me is that you mention net energy flux variations, but not that these variations change the heat content of water and therefore the sea level. Freshwater variations, or net precipitation, is then related to the mass flux. Would make the text clearer and more reader-friendly to be specific.

**Answer:** We have added the following sentences to the related parts in the paragraph.

*"The possible BANOS-attributed heat flux storage can cause thermosteric rise (volumetric) in the sea level."*

*"Accordingly, freshwater flux contribution would indicate the significant effect of BANOS related net precipitation mass variation in the sea level. "*

- page 14, line 30. '...of sea-level and the SLP fields...'.

**Answer:** The text has been updated accordingly.

- page 15. Sensitivity values should have 95% confidence intervals expressed.

**Answer:** The confidence of sensitivity estimations is not critical here, since the sensitivity value has not been used for the estimation of atmosphere-driven sea level change for a given period. By implementing sensitivity estimation, we only intended to compare the responses of the sea level in different locations in this region. The strength of the relation is already described in the correlation analysis, and suggests that the linear relation between sea level variability and BANOS-driven inverse barometric effect strongly covary. Therefore we think that it is not necessary to express the significance level of sensitivity estimations in the manuscript.

However, as it is suggested by the reviewer, we here show the standard error of estimated sensitivity values at the 95% confidence level. The table below shows the sensitivity estimations and corresponding standard error for the IBE contribution estimations.

| Season | Sensitivity Value | Standard err. (±) (95%) |
|--------|-------------------|-------------------------|
| Winter | 18.1 mm/hPa | 4.2 mm/hPa |
|        | 3.44 hPa/u | 0.24 hPa/u |
|        | 62.2 mm/u | 1.0 mm/u |
| Summer | 7.0 mm/hPa | 5.4 mm/hPa |
|        | 1.39 hPa/u | 0.37 hPa/u |
|        | 9.8 mm/u | 2.0 mm/u |

- page 15, lines 15-16. I do not understand this sentence. Could it just be removed?

**Answer:** We have removed that sentence from the text.

- page 16, lines 1-5. I do not understand why a negative (NEF,BANOS) correlation in summer implies that NEF are not responsible for the sea-level variations, while in winter they are. Wouldn't the NEF effect in summer just be the opposite to the winter one? In winter the positive BANOS is associated with increased gain of heat by the sea which contributes to increased sea-level due to the heat expansion. In summer the positive BANOS is associated with increased loss of heat by the sea, one could think of relatively cold air being advected from the Atlantic to the Baltic Sea, which then contributes negatively to sea-level. But the effect is small compared to other factors, as you found.

**Answer:** The negative correlation between NEF and BANOS implies that the effect of heat flux cannot be the main factor explaining sea-level variability in summertime. The negative correlation indicates that when more energy is stored in water, sea level tends to fall occurs in that region, which is physically not possible.  Heat flux, as the reviewer suggests, may be involved in the sea-level variations, but it cannot be the main explaining factor.

- page 17, line 17. What is 'this contribution'? By BANOS?

**Answer:** The sentence has been clarified and modified as the following.

Old version: *"However, this contribution is very small for the North Sea."*

New version: *"However, contribution of the BANOS-related freshwater flux to sea level variation is very small for the North Sea."*

- page 17, lines 21-22. Basin-wide averaging of the freshwater flux may result in a weak BANOS link because precipitation is a rather local phenomena. Some localities seem show rather high correlations adjacent to regions of low correlations. Would the averaging explain the weak winter contribution?

**Answer:** Here the idea is that basins are the water shed areas which charge/discharge the net precipitation (P-E) into/out of the Baltic Sea and/or the North Sea. Therefore, we first showed the correlation pattern between possible driving factor and BANOS-related sea level variation over those basins. Then considering the correlation pattern between the BANOS-index and freshwater flux in wintertime, we have seen a correlation pattern which indicates a rather negligible contribution of the freshwater flux to the BANOS-attributed sea level variability in the Baltic Sea and the North Sea. It should be noted that the correlation pattern between the freshwater flux and the BANOS-index indicates quite strong correlation values in some locals and overall good connection over the Baltic basin in summertime; even in that case only 27% of BANOS-attributed sea level could be explained by the freshwater flux effect.  Considering all these findings, we did not take that effect into account for wintertime.

- page 18, line 24. '...flow more parallel...'.

**Answer:** The text has been corrected accordingly.

- page 18, line 25. 'west-east' should be 'south-east' as the wind vectors in Figure 17?

**Answer:** What we are aiming to say is that if there were westerly (west-east direction) geostrophic winds attributed to the BANOS index, we would have expected a significant water movement from the North Sea to the Baltic Sea basin due to the Ekman transport. But that was not the case.

We have slightly modified the sentence in order to clarify the meaning.

It now reads: *"Even in that case, an atmospheric circulation pattern more efficiently driving higher Baltic Sea level should have presented a geostrophic wind anomalies more oriented in the west-east direction."*

- page 18, lines 22-26. Would a better way to think about the wind contribution be to assume that the BANOS-related winds as anomalies modifying the average climatological atmospheric wind field? In that case, one could argue that is possible that certain BANOS-conditions may assist the water to flow from the North Sea to the Baltic Sea through the Danish Straits. This seems plausible when looking at the Ekman transport vectors in Figure 17.

**Answer:** The usual reasoning in this type of analysis is to link the anomalies of the different variables being considered, in this case anomalous wind and anomalous sea-level. The introduction of the mean wind conditions would require also the consideration of the mean sea-level and complicate the interpretation. The reviewer is right that from the physical point of view it is also reasonable to consider the full physical fields, but the reasoning in terms of deviations from the mean conditions is permissible and common assuming that those deviations are small and linearly related.

- page 18, lines 30-31. Add 'thermal expansion' for clarity.

**Answer:** The text has been updated accordingly.

- page 18, lines 29-32. This is an existing mechanism of coastal sea-level increase, but it could be explained more clearly. Now it is very hard to understand. Are you pointing to the possibility that deep water column expand laterally more effectively than the shallower ones due to the pressure dependence of the sea-water equation of state?

**Answer:** What we mean here is that, under the spatially uniform warming over the off-shore and coastal areas, the amount of thermally expanded off-shore (deep) water would be more than of the shallow water. That extra amount would be transferred to the shallow areas, which would, at the end, increase the sea level by distributing a certain amount of water mass over the coastal areas.

We now have modified that part in the manuscript.

Old version: *"Coastal sea-level can also be affected by off-shore sea-level variations in the open ocean. For instance, under spatially uniform warming, sea-level at the coast will rise more than corresponding to its smaller thermal expansion of its shallower water column due to the partial*

*transfer of water mass from the open ocean caused by the stronger expansion of its deeper water column"*

New version: "*Coastal sea-level can also be affected by off-shore sea-level variations in the open ocean. For instance, under spatially uniform warming, sea-level at the coast will rise more than corresponding to its smaller thermal expansion of its shallower water column due to the partial transfer of water mass from the open ocean. Since the deep ocean water is expected to have stronger expansion than the shallow water due to its deeper water column.*"

- page 19, You could write about potential applications using the BANOS index. This would promote its use among the readers.

**Answer:** The following part has been added to the Section 5:

[revised manuscript text omitted]